# SE(3)-equivariant ternary complex prediction towards target protein degradation

Fanglei Xue [1], Meihan Zhang [2], Shuqi Li [3], Xinyu Gao[4], James A. Wohlschlegel [5], Wenbing Huang [3,6] ✉, Yi Yang [7] ✉ & Weixian Deng [5] ✉

Targeted protein degradation (TPD) has rapidly emerged as a powerful modality for drugging previously "undruggable" proteins. TPD employs small molecules like PROTACs and molecular glue degraders (MGD) to induce target protein degradation via the formation of a ternary complex with an E3 ligase. However, the rational design of these degraders is severely hindered by the difficulty of obtaining these ternary structures. Here we introduce Deep-Ternary, a novel end-to-end deep learning approach using an SE(3)-equivariant encoder and a query-based decoder to accurately and rapidly predict these critical structures. Trained on carefully curated TernaryDB, DeepTernary achieves state-of-the-art performance on PROTAC benchmarks without prior exposure to known PROTACs and shows notable prediction capability on the more challenging MGD benchmark with a blind docking protocol. Remarkably, the buried surface areas calculated from predicted structures correlate with experimental degradation potency metrics. Overall, DeepTernary offers a powerful tool for the development of targeted protein degraders.

Targeted protein degradation (TPD) is a rapidly evolving field in drug discovery, representing a promising therapeutic approach to degrade target proteins via harnessing the ubiquitin-proteasome system and autophagy-lysosome system[1–4]. Traditional drug discovery mainly focuses on inhibiting the activity of target proteins, which may not always be effective, especially in cases where the target protein is 'undruggable' by occupancy-driven inhibitors like small molecules[5]. These "undruggable" proteins include oncology targets in the SWI/SNF complex[6,7] and many kinases[8] which share high homology active domain with their essential non-disease related family members, and transcriptional factors[9] that are highly unstructured until they form active conformations. TPD presents an alternative strategy, which is to induce the degradation of target proteins rather than inhibit their activity to achieve desirable therapeutic outcomes. The mode of action (MOA) for TPD offers several advantages: Firstly, TPD molecules do not require targeting "active site", allowing them to selectively target disease-driver proteins without affecting other essential homologous proteins that often share conserved active sites, and exert potential to engage highly-unstructured transcriptional factors[10] and other scaffolding targets that do not depend on active sites[10]. Secondly, its transient protein interaction via event-driven mechanism reduces the reliance on strong binding affinity, in contrast to inhibitor drugs[11]. Furthermore, its catalytic nature mitigates the requirement for high dosages and the subsequent challenges associated with off-target effects[12]. Lastly, even for existing targetable proteins by inhibitors, it still offers

[1]ReLER Lab, AAII, University of Technology Sydney, Sydney, NSW 2007, Australia. [2]College of Life Sciences, Nankai University, Tianjin, China. [3]Gaoling School of Artificial Intelligence, Renmin University of China, Beijing, China. [4]University of Chinese Academy of Sciences, Beijing, China. [5]Department of Biological Chemistry at David Geffen School of Medicine, University of California, Los Angeles, Los Angeles, CA, USA. [6]Beijing Key Laboratory of Research on Large Models and Intelligent Governance, Beijing, China. [7]ReLER Lab, CCAI, Zhejiang University, Hangzhou, China. ✉e-mail: hwenbing@ruc.edu.cn; yangyics@zju.edu.cn; weixiandeng@ucla.edu

alternative therapeutic options to fight against drug resistance caused by active site mutations[13].

Proteolysis-targeting chimeras (PROTACs) and molecular glue degraders (MGDs) are two main modes of TPD[11]. As shown in Fig. 1a, PROTACs are hetero-bifunctional small molecules consisting of three moieties, including a warhead, which is the ligand of the protein of interest (POI), an anchor, which is the ligand of an E3 ubiquitin ligase being employed, and a linker linking the warhead and anchor. With the hetero-bifunctional structure, PROTACs recruit the POI to an E3 ubiquitin ligase, leading to the ubiquitination of the POI and its subsequent degradation process by UPS[2,14]. As of January 2023, there have been 18 PROTACs under evaluation by regulatory authorities, targeting different malignant cancer diseases[15]. MGDs, in contrast, are small molecules that facilitate the interaction between the POI and an E3 ubiquitin ligase, enabling the ubiquitination and degradation processes of the POI[16]. Unlike PROTACs, they do not require a bifunctional structure but act by stabilizing existing protein–protein interactions or inducing new interactions[17]. Despite their distinct modes of action, both PROTACs and MG(D)s share a common feature: the induction of a ternary complex structure that is crucial for their respective mechanisms.

Understanding the ternary structure induced by PROTACs or MGDs provides crucial insights into the molecular basis of induced protein degradation. In the context of PROTACs, the ternary structure

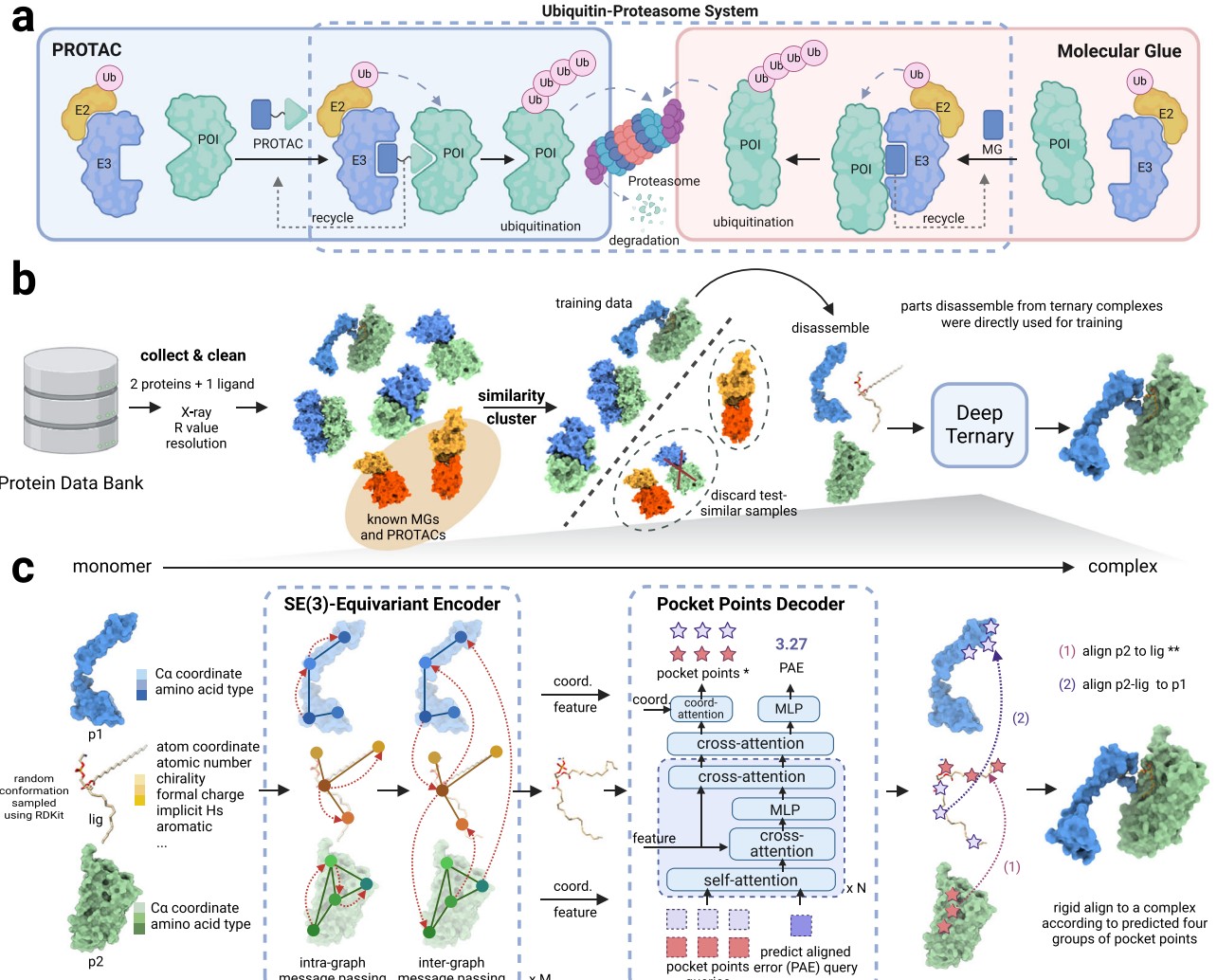

**Fig. 1 | DeepTernary is a deep learning model for predicting the structure of the ternary complex induced by PROTACs and MG(D)s. a** The MOA of PROTACs and MGDs. The protein of interest (POI) and the E3 ligase are recruited to proximity by PROTACs or MGDs to form a ternary complex, and then the Ubiquitin-Proteasome System (UPS) is employed to transfer the ubiquitin and degrade the POI. **b** To mitigate the scarcity of known PROTACs and MG(D)s structures, a large-scale ternary complex dataset (named TernaryDB) was collected by searching and cleaning complexes from the Protein Data Bank (PDB) archive. The collected samples were then grouped into clusters by similarity. Any complex that is similar to known PROTAC and MG(D) induced complexes was excluded from the training set. DeepTernary was trained on this filtered database by predicting the original complex structure using disassembled monomers. **c** DeepTernary is an SE(3)-equivalent graph neural network equipped with attention blocks to facilitate efficient information exchange. It begins by representing two proteins and a small molecule as three graphs, encoding node coordinates, diverse amino acid or atom characteristics as node features, edge types, and distances as edge features. The three graphs are fed into an encoder consisting of a series of SE(3)-equivariant blocks, enabling both intra- and inter-graph learning to capture interactions effectively. The encoder will predict the conformation of the small molecule and output the refined node features/coordinates of the two proteins. Subsequently, a decoder comprising several attention-based blocks employs these refined features/coordinates to generate two pairs of pocket points and a predicted aligned error (PAE). The pocket points are then used to align both the small molecule and protein2 to protein1. * For PROTAC, the pocket points are derived from unbound structures, don't need to be predicted. ** For MG(D), the ligand and protein2 are simultaneously aligned to protein1. Image created with BioRender. Xue, F. (2025) https://BioRender.com/o91e3ly, with permission.

elucidates how the PROTAC molecule facilitates the connection between the POI and the E3 ligase, demonstrating the interacting poses, properties of the contact interface, and solvent-exposed amino acid residues essential for efficient ubiquitination. For instance, the buried surface area (BSA) of the ternary structure[18] is a critical parameter indicating the extent of interaction surface between the PROTAC, the POI, and the E3 ligase, directly correlating with the stability and efficacy of the induced degradation[17,19]. The ternary structure can also suggest possible modifications in terms of the length and the composition of the PROTAC linker in order to improve selectivity and reduce off-target effects[19]. Similar to PROTACs, the BSA of the MGD-induced ternary structure is a crucial determinant of their functional impact, influencing both the strength and specificity of the interaction between the POI and the E3 ligase[17], which can also provide clues about the molecular features that are crucial for the molecular glue's activity[20].

Existing experimental approaches to obtain the PROTAC- or MGD-induced ternary structures, such as X-ray crystallography and cryo-EM, often depend on costly instrumentation and intricate reagents and remain a formidable challenge for seasoned structural biologists due to the necessity of high-purity proteins and precise buffer conditions. Instead, in silico approaches have been proposed to predict ternary structures that primarily using various docking methods (such as PatchDock[21,22], FRODock[23,24], RosettaDock[22,24,25], and PIPER[19,26,27]) to generate big pools of structures and then to rank, filter, and refine the docked ternary structures by minimizing free energy[19,22,27], atom clash[27–29], constraining distance to E2 ligase[27], and molecular dynamics simulations[19]. In spite of the encouraging progress, the structures predicted by existing docking methods still deviate greatly from experimentally determined ones, and the docking process is usually time-consuming. Recently, deep learning technologies such as AlphaFold2[30] and RosettaFold[31] have shown promising prediction accuracy for protein structure prediction by making use of deep and sophisticated neural networks to distill crucial features from extensive training datasets. These remarkable achievements have attracted significant scientific interest in extending deep learning to other related tasks, including protein–protein[32,33] and protein–ligand complex structure prediction[34,35]. However, to our best knowledge, there were no reported research on predicting PROTAC- or MGD-induced ternary structures by using deep learning approaches. This can be attributed to the heightened complexity of modeling ternary structures compared to the unitary or binary structures tackled in prior studies. Additionally, the scarcity of training data presents a significant obstacle to training deep learning models, as there are only a few resolved ternary structures for both PROTACs and MGDs[17,36], making it impractical to train such models with such limited data.

In this work, we introduce a deep learning-based framework for predicting ternary complexes induced by PROTACs and MG(D)s (We use the term MG(D) to denote both degraders and non-degraders, as the formation of MGD ternary complexes can be generalized to non-degraders). To achieve this, we curated TernaryDB, a large-scale dataset comprising over 20,000 ternary complexes from the protein data bank (PDB). The dataset focuses on high-quality complexes that include a small molecule and two proteins while deliberately excluding known PROTACs and MG(D)s from the training list. Figure 1b outlines the construction process of the dataset. Leveraging TernaryDB, we trained DeepTernary, an SE(3)-equivariant graph neural network specifically designed for ternary structure prediction (Fig. 1c). In this model, the ternary complexes were disassembled into three components-p1 (protein1), lig (ligand), and p2 (protein2)-each modeled as a graph. Graph neural networks(GNN)[37] offer a powerful framework for processing graph-structured data through message passing between nodes and edges. To improve data efficiency, we employed an SE(3)-equivariant GNN, leveraging the symmetry properties of SE(3) to ensure invariance to the translation and rotation of 3D

structures. Additionally, we introduced a novel ternary inter-graph attention mechanism to capture the intricate relationships between ternary components, along with a query-based pocket points decoder to predict the final complex structure. With these innovations, DeepTernary effectively predicts both the conformation of the small molecule and the docking poses of the ternary complex. DeepTernary was evaluated against existing PROTAC and MG(D) benchmarks, achieving state-of-the-art performance with DockQ scores of 0.65 and 0.21, with average inference times of ~7 and 1 s, respectively. The model's ability to generalize from a non-PROTAC/MG(D) PDB dataset to PROTAC/MG(D) ternary structures highlights its capacity to capture the fundamental interaction patterns governing ternary complex formation, rather than relying on memorization. Moreover, the predicted buried surface area (BSA) of the PROTAC complexes aligns closely with values reported in existing literature, with BSA ranging from 1100 to 1500, indicating high degradation potential. These results demonstrate DeepTernary's potential to advance our understanding and manipulation of protein degradation mechanisms.

## Results
### The construction of TernaryDB
There are only a few dozen experimentally determined PROTAC- and MG(D)-involved ternary complexes in the PDB[38]. Despite the remarkable success of deep learning in protein structure prediction[30,31], protein–protein docking[32], and protein–ligand interactions[34,39], its application to targeted protein degradation (TPD) remains underdeveloped, primarily due to the scarcity of training data. We hypothesized that TPD complexes adhere to the same fundamental atom-interaction principles as other tripartite complexes. To test this hypothesis and enable deep learning-based prediction of TPD complex structures, we curated a comprehensive dataset of ternary complexes from the PDB. After stringent data filtering (details are provided in Methods 4), the final dataset comprised 22,303 complexes, with their key attributes illustrated in Fig. 2. The distribution of ligand atom counts, excluding hydrogens, is shown in Fig. 2b, revealing that the majority of ligands contain fewer than 60 heavy atoms, with only a small subset exceeding 100. The chemical diversity of these ligands, represented by Morgan fingerprints (Fig. 2f), highlights the broad chemical space and drug-like properties of the dataset. Proteins from 363 species, ranging from bacteria to humans, are included in the dataset (Fig. 2d). Although the protein space is relatively sparse, it adequately covers PROTAC- and MG(D)-induced proteins (Fig. 2e).

To rigorously assess our method, we integrated known PROTAC and MG(D) ternary complexes into the test sets. To prevent data leakage, we utilized MMseqs2[40] to cluster the dataset based on protein sequence similarity. Clusters containing known PROTAC or MG(D) complexes were excluded from the training set and served as a validation set if their ligands were not similar to the test set (detailed in the Methods section), ensuring no overlap between training and test data. This clustering approach yielded 16,203 complexes distributed across 1398 clusters for PROTACs and 22,046 complexes across 1982 clusters for MG(D)s. The distribution of cluster sizes is shown in Fig. 2c, where most clusters are small, although a few contain over 100 complexes.

To mitigate potential biases during training, we adopted a cluster-wise sampling strategy. Traditional uniform sampling within batches could result in the selection of highly similar complexes, thereby skewing the training process. Instead, we first randomly sampled a cluster with equal probability and then selected the representative complex with a 20% likelihood; otherwise, a random complex from the cluster was chosen. The representative complex was determined using the MMseqs2 toolkit during clustering. This approach ensures a diverse and representative sampling of the training data, enhancing the model's ability to generalize across complex structures.

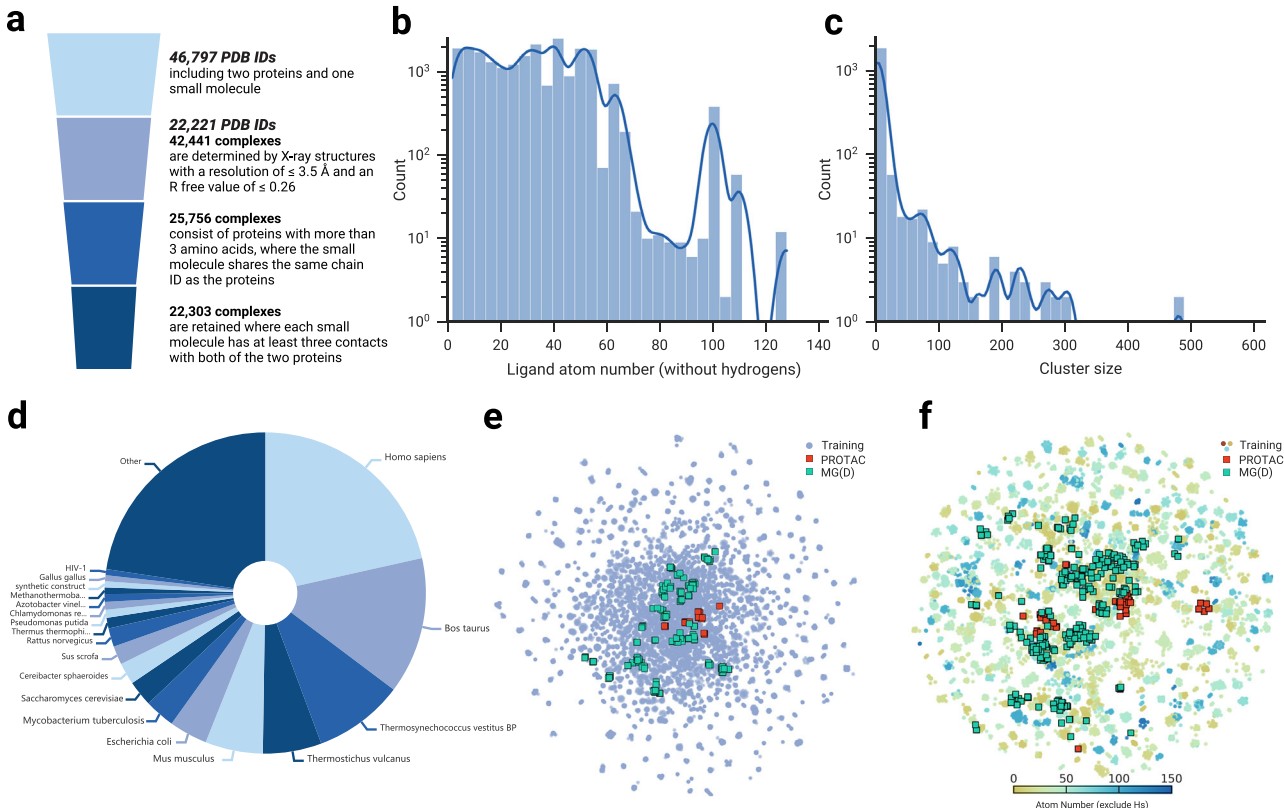

**Fig. 2 | TernaryDB construction and visualization. a** The process of collecting and cleaning the ternary complexes dataset. Initially, a search of ternary structures from the PDB yielded 46,797 PDB IDs, each of which contains at least two proteins and one small molecule. High-quality PDB IDs were retained based on criteria such as X-ray crystallography data, resolution, and R-free value. From this subset, 42,441 complexes were extracted, each comprising just two proteins and one small molecule. These complexes underwent further refinement based on peptide chain length and the number of contacts. Ultimately, 22,303 complexes met our stringent criteria and were used to train our model. **b** Histogram of the ligand atom number (excluding hydrogens) within the dataset. **c** Histogram of cluster sizes within the dataset according to the protein sequence similarity. **d** The distribution of protein

source organisms in the dataset. **e** Proteome-wide view of the collected dataset. ESM-1b[60] sequence embeddings for the two proteins in each complex are calculated and concatenated. This is followed by two-dimensional (2D) uniform manifold approximation and projection (UMAP). Similar complexes to PROTACs- and MG(D)s-involved ternary structures are denoted as red and green square points, respectively. **f** Chemical space covered by the dataset. Morgan fingerprints are converted to 1024-length vectors and visualized through a 2D UMAP. The points on the map are differentiated and colored by molecular weight (hydrogen excluded). PROTACs- and MG(D)s-like molecules are highlighted as red and green square points, respectively.

## The architecture of DeepTernary

DeepTernary is designed to predict the structures of small molecule-induced ternary complexes, such as those formed by PROTACs and MG(D)s-induced E3 ligase with POI complexes. Unlike existing methods that rely on standard protein–protein docking programs to approximate the interaction between two proteins, often neglecting the presence of small molecules, DeepTernary employs a deep neural network to directly learn the intricate dynamics of protein–protein and protein–ligand interactions within ternary complexes. As shown in Fig. 1b, the model is trained by learning to re-dock disassembled ternary structures. During inference, for PROTAC-induced complexes, DeepTernary utilizes the respective monomer forms of the two protein structures (E3 ligase and target protein) along with docked warheads and anchors from other PDB entries (unbound structures), in addition to the PROTAC Simplified Molecular Input Line Entry System (SMILES) strings. For MG(D)-induced complexes, where obtaining unbound structures poses a challenge, we instead employ the in-complex form of the two protein structures (subjected to random rotations and transformations) and with the corresponding MG(D) SMILES strings as input.

Building upon these input modalities, the overall architecture of DeepTernary is composed of an encoder and a decoder that work in tandem to capture and reconstruct the intricate structural details of the ternary complexes. The process begins by generating a random conformation of the small molecule using RDKiT[41] and randomly displacing

the small molecule and protein2 (p2) away from protein1 (p1). This serves as the starting point for learning the interactions between the two proteins and the ligand. As illustrated in Fig. 1c, these three monomers are encoded as graphs and processed through an SE(3)-equivariant encoder. This encoder facilitates the interaction of the encoded entities in a geometrically consistent manner. Multiple blocks of alternating intra- and inter-graph message passing are employed to update the coordinates and latent features of the three monomers. To efficiently capture the symmetry in their interactions, the parameters in the encoders for p1 and p2 are weight-shared (Method 4). Following the encoding stage, the confirmation of the small molecule is utilized as the final conformation. The final ternary structure is generated based on this predicted conformation and pocket points. For PROTACs, the pocket points are derived from unbound structures, while for MG(D)s, these points are predicted by the proposed prompt-based pocket points decoder (PPPD). With this information, we can rigidly align the ligand and p2 back to p1 to form the final structure. The PPPD will also predict an alignment error for this predicted structure. Notably, benefiting from the Transformer architecture's inherent ability to handle variable numbers of input queries without architectural modifications, the proposed PPPD architecture is unified for both PROTAC and MG(D). This simplifies the model design and implementation. For PROTACs, only PAE queries are input to the decoder, while for MG(D)s, both pocket point and PAE queries are used.

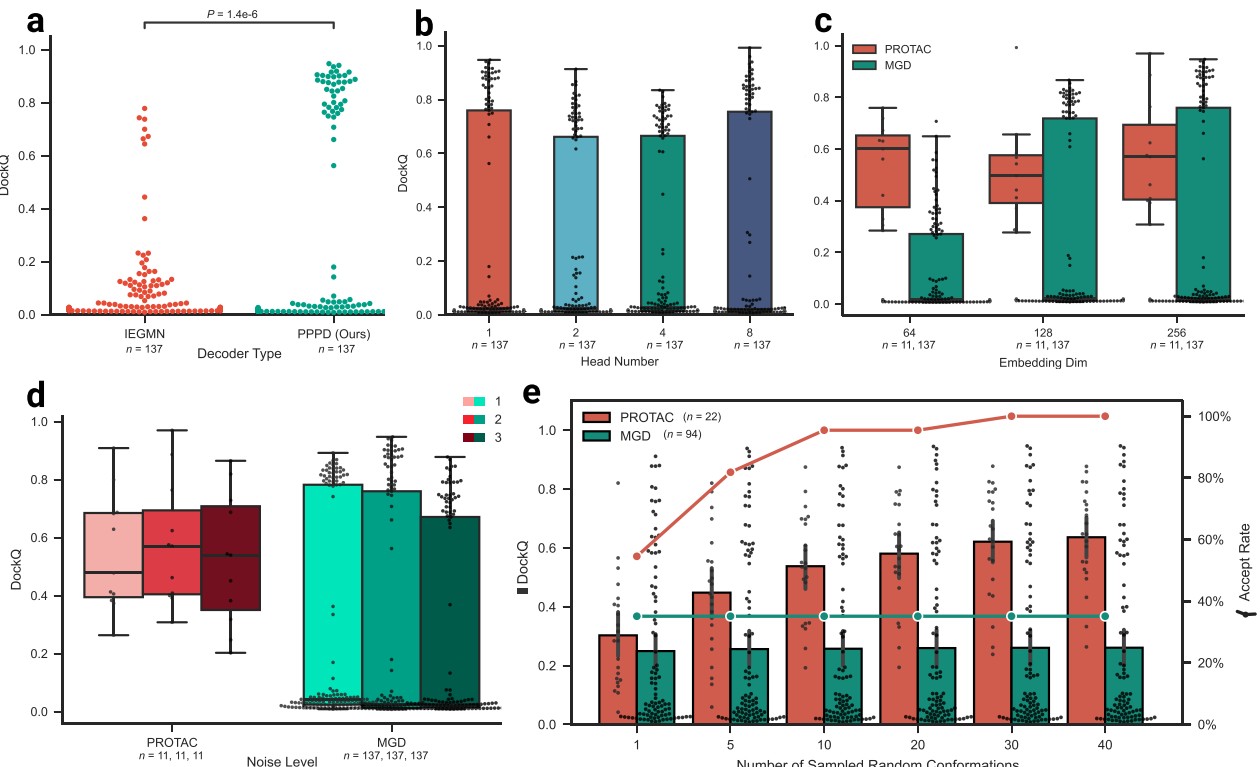

**Fig. 3 | Effectiveness of DeepTernary designs on PROTAC and MG(D) validation benchmarks. a** Comparison of decoder types: Our proposed pocket points decoder significantly outperforms IEGMN in predicting more medium- to high-quality binding poses (DockQ >0.49). Statistical analysis was performed using the two-sided independent $t$-test. **b** Impact of multi-head attention on coordination prediction: A single attention head achieves performance comparable to an eight-head configuration, while reducing computational demands. **c** Influence of latent embedding dimension on model performance: VPS results indicate that larger dimensions enhance learning learning, especially for MG(D) complexes. **d** Effect of noise level on model robustness: Increasing the noise level from 1 to 2 improves performance on both PROTAC and MG(D) benchmarks, as reflected by VPS.

**e** Impact of the number of sampled random conformations on test results: An increased number of sampled conformations correlates with higher DockQ scores and acceptance rates (DockQ >0.23) for PROTACs, while MG(D) performance remains largely unchanged. Data are presented as mean values with error bars representing 95% confidence intervals estimated from 22 PROTAC and 94 MG(D) samples. (All box plots in this figure extend from the first quartile (Q1) to the third quartile (Q3) of the data, with a line at the median. The whiskers extend from the box to the farthest data point lying within 1.5x the inter-quartile range (IQR) from the box. Flier points are those past the end of the whiskers. The sample number ($n$) is annotated under each plot. Source data are provided as a Source Data file.

## Effectiveness of model designs

Based on the binary interaction prediction model[32,34], we had explored various choices of model designs and hyperparameters for Deep-Ternary. To ensure robust model selection, we employed a validation set consisting of curated structures that are dissimilar to the training set and also not similar to the test set (ligand Tanimoto similarity <0.85). Model performance on this validation set was assessed using a simple score calculated as the average of the DockQ scores for the top-ranked prediction and the best overall prediction (detailed information is provided in Supplementary Sec. 1.1).

First, DeepTernary incorporates a ternary inter-graph attention mechanism in the encoder, enabling it to capture more complex ternary interactions. However, the initial decoder design, denoted as IEGMN, struggled to effectively translate the encoded information into accurate binding poses (Fig. 3a). By introducing the newly developed Prompt-based Pocket Points Decoder (PPPD) (detailed in Methods 4), we significantly enhanced performance, with many samples achieving medium to high quality (DockQ >0.49). Additionally, although multi-head attention is advantageous in natural language processing, it performs less effectively in predicting pocket point coordinates for our model. Specifically, increasing the number of attention heads initially causes a slight decrease in the DockQ score, which then recovers when eight heads are used (Fig. 3b). Therefore, we adopted single-head

attention in the PPPD to achieve accurate and efficient coordinate extraction. Transitioning from binary to ternary interaction prediction posed additional challenges. We discovered that increasing the latent embedding space improved the model's capacity to learn complex triplet interactions, particularly for MG(D) complexes, which exhibit greater structural complexity (Fig. 3c). Besides, to avoid the risk of overfitting, we increased the noise added to both the coordinates and latent features, from 1 to 2, which improved performance across both PROTAC and MG(D) benchmarks (Fig. 3d). Nevertheless, adding too much noise (noise level from 2 to 3) will hinder the performance.

In line with previous studies[24,27,36], DeepTernary utilizes RDKit[41] to generate initial conformations for small molecules, sampling multiple conformations with different seed numbers during inference. Our ablation studies (Fig. 3e) demonstrated that both the DockQ score and accept rate (DockQ >0.23) increased as the number of sampled conformations for PROTAC grew. Conversely, MG(D) complexes showed little change. This discrepancy can be attributed to the fact that PROTACs have more atoms and exhibit greater structural flexibility, while MG(D)s have a smaller conformation space. Based on these findings, we sample 40 initial random conformations for each PROTAC and rank the predicted results using the PAE score. For MG(D) predictions, we use a single initial conformation to conserve computational resources.

## DeepTernary achieves the highest accuracy in PROTACs-induced ternary structure prediction

To evaluate our method, we utilized the PROTAC benchmark compiled by ref. 36, which consists of 22 known PROTAC-induced ternary structures serving as the test set. The unbound protocol adopted in this benchmark emulates the real-world scenario encountered during drug discovery, where the experimental structure of the ternary complex is often unavailable. In this protocol, an unbound complex refers to a protein with a bound ligand similar to the warhead or anchor of the PROTAC, but not co-crystallized with the entire PROTAC molecule. To align with the rational design process of PROTACs, we followed this unbound protocol to evaluate DeepTernary.

To mitigate data leakage, we excluded any similar protein pairs from the dataset used to train our model. Unlike previous methods, which rely on human-defined heuristics—such as manually set thresholds for free energy[19,22,27], atom clashes[27–29], or linker ends distances[27]—to filter ternary conformations, we leveraged deep learning to automatically capture high-dimensional interactions between PROTACs and proteins. Furthermore, in contrast to refs. 28,29, our model was trained without any PROTAC-involved structures and directly evaluated on the PROTAC benchmark, using a zero-shot protocol. This approach tests the model's ability to learn general interaction rules applicable to any ternary structure, not just those induced by PROTACs, offering a stringent measure of how well the model generalizes from non-PROTAC to PROTAC data.

For a comprehensive comparison, we employed several evaluation metrics, including DockQ scores (Fig. 4a), rank of the first prediction achieving a DockQ score greater than 0.23 (Fig. 4b), the percentage of CAPRI high/medium/acceptable predictions (Fig. 4d), and the percentage of predictions with RMSD <10 Å (Fig. 4f). These metrics are detailed in Evaluation metrics. As we can see, DeepTernary consistently produces higher DockQ scores and higher rates of acceptable predictions (both in terms of High/Medium/Acceptable predictions and <10 Å) compared to other published methods, including FRODock- and RosettaDock-Based methods[24], BOTCP[36], Method 4[28], Method 4B[29], PRosettaC[22], and most recently published AlphaFold3[42] and Chai-1[43]. Specifically, it achieved an average DockQ score of 0.65 across the test set, significantly outperforming the recently proposed BOTCP[36], which scored 0.44. Although other methods were only evaluated on subsets of the benchmark, DeepTernary demonstrated superior performance across overlapping tested structures. Notably, as illustrated in Fig. 4c, DeepTernary surpasses the top-performing RosettaDock-based method[24] for most testing structures.

PROTAC molecules, with their larger atom counts compared to natural small molecules, exhibit diverse conformations due to their significant degrees of freedom. To model this flexibility, we employed the RDKit toolkit to generate multiple initial conformations of the ligand using different random seeds, each of which was input into our model. To estimate the prediction quality, we introduced a predicted aligned error (PAE), allowing us to rank the predicted results and select the most confident output. With an average rank of 4.06 under 40 seeds (Fig. 4b), DeepTernary reliably generated acceptable predictions (DockQ >0.23). In other words, there is generally at least one acceptable prediction within the top four results. To compare with existing methods, we also calculate the prediction success rate for each complex based on another two criteria: CAPRI criteria and RMSD <10 Å (Fig. 4d, f). As we can see, DeepTernary significantly improves the success rate to around 50%, which means for most of the test complexes, more than half of the predictions are above acceptable quality. Since the ground-truth structure is typically unavailable in practice, distinguishing between higher- and lower-quality output structures remains challenging without a reliable scoring or ranking system. Our DeepTernary addresses this challenge by incorporating a PAE predictor, where lower PAE values indicate higher confidence in predictions (Supplementary Fig. 2). The mean Top-1 DockQ based on PAE

reaches up to 0.4 (Supplementary Table 5), surpassing the acceptable cutoff of 0.23, which enhances its utility in real-world drug discovery applications.

Finally, we examined the performance of DeepTernary across the three distinct E3 ligases present in the 22 benchmark complexes. As shown in Fig. 4d, DeepTernary consistently achieved desirable DockQ scores across all ligases, highlighting its robustness and generalizability. Visual comparisons of the predicted and experimentally determined structures (Fig. 4e, f) demonstrate that our model can generate high-quality predictions, with DockQ values exceeding 0.9. Notably, for PDB IDs 6W7O and 6W8I, which share the same E3 ligase and POI pair but differ in their PROTACs, DeepTernary accurately captured the structural differences, producing predictions aligned with experimental expectations.

## DeepTernary reaches acceptable accuracy in MG(D)-induced ternary complex structure prediction

Molecular glue degraders (MGDs) represent a novel class of TPD drugs, distinct from PROTACs due to their lower molecular weight and alternative MOA. These characteristics often result in an advantageous starting point for medicinal chemistry optimization, as well as enhanced drug-like physicochemical properties[44]. Their simplicity in structure further facilitates later-stage drug development. The rising interest in MGDs has prompted significant research efforts and corporate investments focused on this new modality. In particular, structure-based rational design plays a crucial role in maximizing the chances of successful drug discovery. For instance, the crystal structure of the $\beta$-TrCP, $\beta$-catenin, and NRX-1933 ternary complex has been instrumental in developing MGDs with improved mutant selectivity[45]. Similarly, the discovery of ALV2, a mutant-specific Ikaros degrader, relied on known crystal structures for guidance[46,47].

MG(D)s can either stabilize endogenous protein–protein interactions or induce non-native ones[48]. However, predicting MG(D)-induced ternary complex structures poses a challenge due to the often weak binding affinity between the small molecule and one of the proteins. With no existing in silico method specifically designed for MG(D)-induced complexes, we employed EquiDock[32], a protein–protein docking approach, to test whether weak interactions between two proteins could approximate MG(D)-induced binding features. Using MG(D)-induced complexes collected by Rui et al.[17] as a test set, we evaluated the models' performance using DockQ scores, similar to our approach with PROTAC experiments. The results, shown in Fig. 5a, reveal that EquiDock achieves an average DockQ score of only 0.04. In contrast, DeepTernary significantly improves the score to 0.21, demonstrating the advantage of incorporating small molecule information and modeling ternary interactions within the model architecture.

Recently, DeepMind introduced AlphaFold3 (AF3)[42], which is able to predict complexes involving nearly all molecular types in the PDB, including proteins and small molecules. However, since the code has not yet been released and the AlphaFold Server does not currently allow customization of small molecules, we used AF3 solely for protein–protein binding predictions, as we did with EquiDock. Figure 5b illustrates a prediction for PDB ID 7BQU, where AF3, thought better than EquiDock, performs significantly worse than DeepTernary, which predicts a structure closest to the co-crystallized ground truth (green vs. gray). We show the PAE matrix in the top right corner. The PAE in AlphaFold, measured in Ångströms (Å), represents the expected positional error between two residues in the predicted structure. Typically, an AF PAE value exceeding 15 Å is considered indicative of a less confident prediction. In this prediction, the PAE values are ~20 Å, highlighting AF3's lower confidence in this predicted interaction.

Rui et al. categorized the collected MG(D)-induced complexes into two groups based on the nature of their protein–protein interface: Group 1 involves domain-domain interactions, where two proteins bind through well-structured domains (as shown in Fig. 5d), and Group

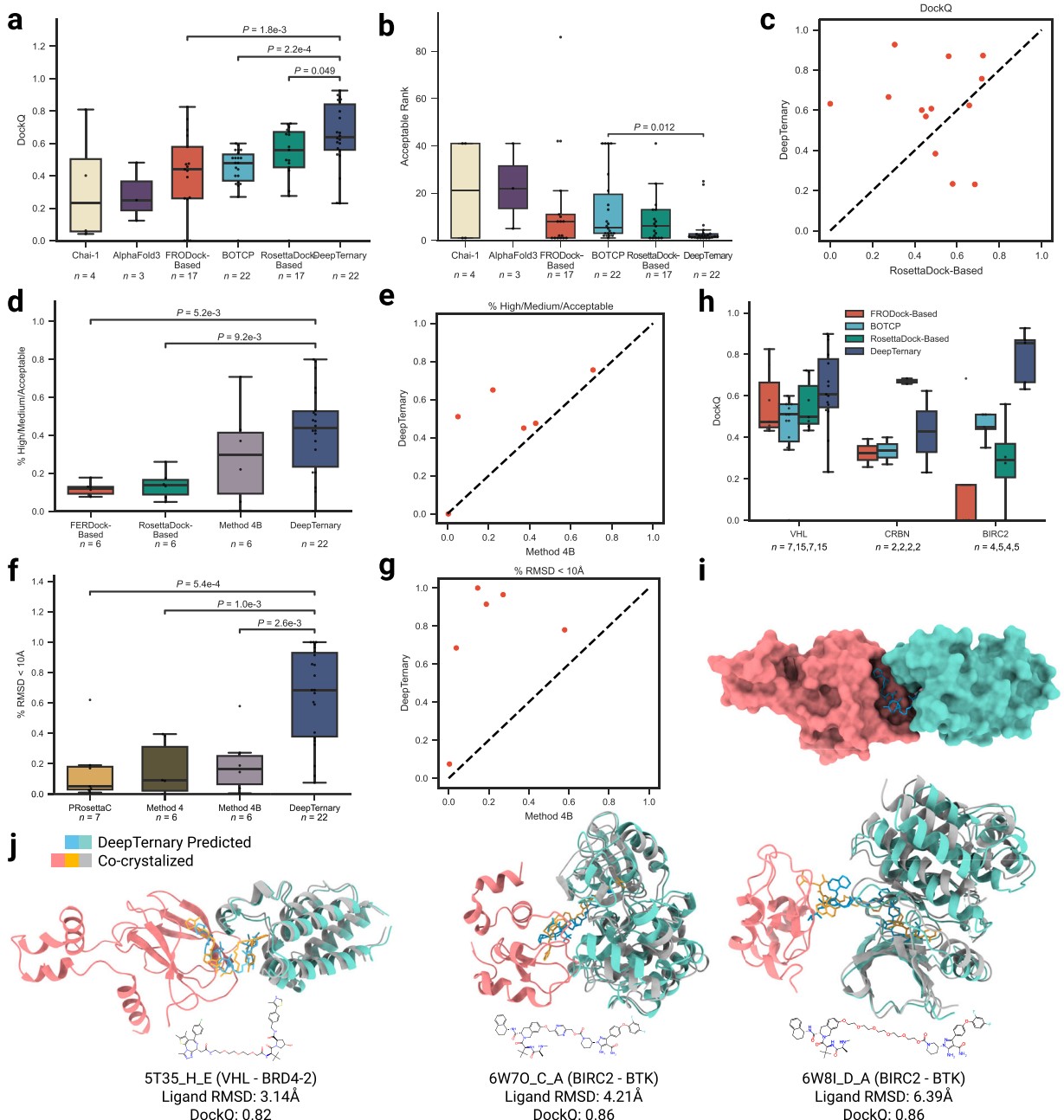

**Fig. 4 | DeepTernary achieves the highest accuracy in PROTACs-induced ternary structure prediction.** DeepTernary outperforms existing methods in predicting 22 existing PROTAC-induced ternary structures regarding the metrics of the DockQ score (**a**) and the first acceptable rank containing at least one prediction with DockQ ≥0.23 (**b**). For those that failed to generate an acceptable result, we manually set the rank value to 41 for a fair comparison. Besides, DeepTernary achieves better DockQ performance on most complexes compared to the current best model, the RosettaDock-Based model (**c**). **d, e** DeepTernary outperforms existing methods on the percentage of High/Medium/Acceptable.
**f, g**, DeepTernary has a higher potential to generate decent (RMSD <10 Å) results. **h** DockQ performance comparison among different E3 ligases. **i** Surface illustration

of the predicted structure of PDB ID 5T35. **j** Three examples of predicted ternary structures (teal and orange for the protein and ligand, respectively) overlaid with the experimental structures (gray and green for the protein and ligand). The receptor protein is colored red, and the chemical structure diagrams of PROTAC molecules are illustrated at the bottom. (All box plots in this figure extend from the first quartile (Q1) to the third quartile (Q3) of the data, with a line at the median. The whiskers extend from the box to the farthest data point lying within 1.5x the interquartile range (IQR) from the box. Flier points are those past the end of the whiskers. Statistical analyses were performed using the two-sided independent *t*-test. The sample number (*n*) is annotated under each plot. Source data are provided as a Source Data file.

2 involves sequence motif-domain interactions, where a protein sequence motif binds to a structured domain (illustrated in Fig. 5c). Our results indicate that both EquiDock and DeepTernary perform better on Group 2 complexes than on Group 1, as shown in Fig. 5e. This suggests that the large, well-folded domains in Group 1 complexes involve more complex binding rules, which may not be adequately covered by the training set (Supplementary Fig. 3). In contrast, the

interactions involving small recognition motifs in Group 2 are better captured, leading to improved predictions.

**The total buried surface area (BSA) from our predicted structures strongly correlates with degradation potency**
Experimental work by ref. 19 has demonstrated a strong correlation between the total buried surface area (BSA) of PROTAC-mediated

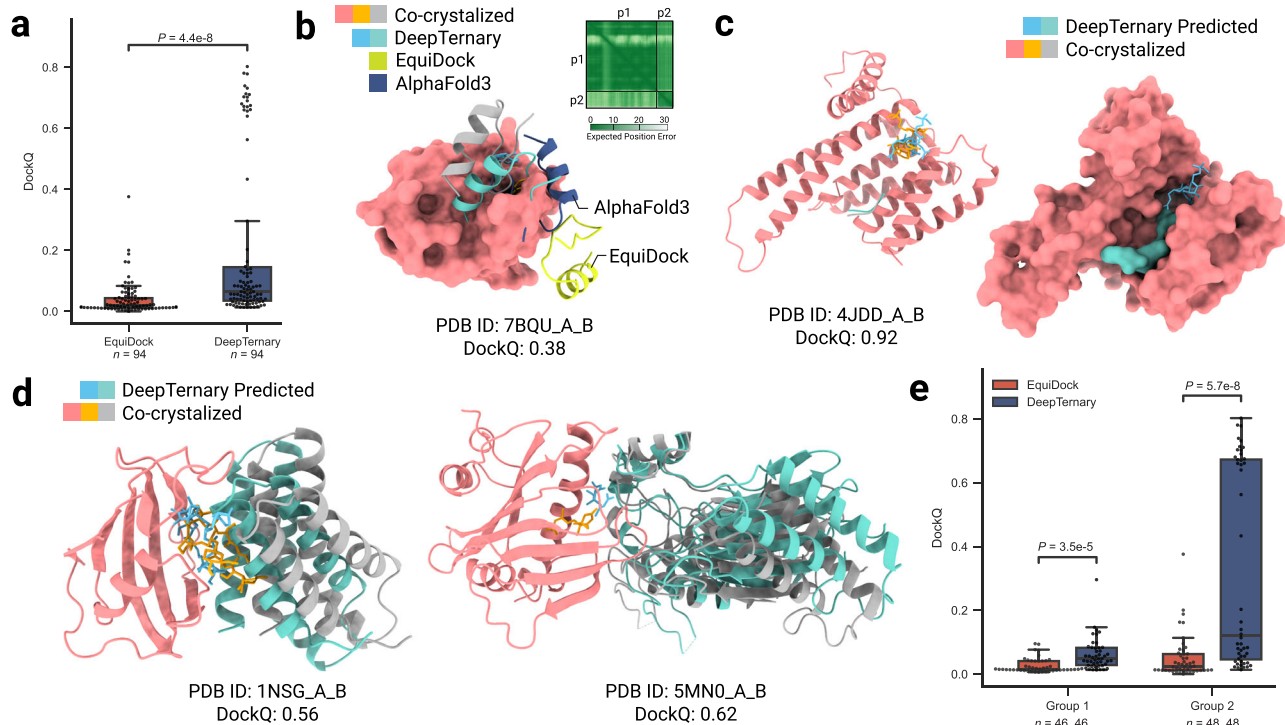

**Fig. 5 | DeepTernary reaches acceptable accuracy for MG(D)-induced ternary complex structure prediction. a** Since no MG(D)-induced complex prediction method exists, we compared DeepTernary with the traditional protein–protein interaction (PPI) prediction method EquiDock. DeepTernary significantly outperforms EquiDock by precisely modeling ternary interactions. **b** Comparison with EquiDock and the recently realeased AlphaFold3 (AF3) on PDB ID 7BQU. The predicted aligned error (PAE) matrix from AF3 is shown in the top right corner. AF3 predicts the structure with plDDT values between 50 and 90 but shows low confidence for docking, with PAE values exceeding 20 between p1 and p2. **c** Visualization of the predicted ternary structure for PDB ID 4JDD (Group 2),

displayed using both cartoon and surface illustrations. **d** Two predicted results from Group 1. **e** Performance comparison across different interaction modes: domain-domain (Group 1) and sequence motif-domain (Group 2). (All box plots in this figure extend from the first quartile (Q1) to the third quartile (Q3) of the data, with a line at the median. The whiskers extend from the box to the farthest data point lying within 1.5x the inter-quartile range (IQR) from the box. Flier points are those past the end of the whiskers. Statistical analyses were performed using the two-sided independent $t$-test. The sample number ($n$) is annotated under each plot. Source data are provided as a Source Data file.

ternary complexes and the equilibrium dissociation constant ($K_{LPT}$) for VHL-PROTAC-SMARCA2. Their findings revealed that BSA has a negative correlation with $ln(K_{LPT})$, while a lower $K_{LPT}$ corresponds to higher degradation potency. In other words, a higher BSA corresponds to higher degradation potency. To test whether our predicted ternary structures could reflect this relationship, we calculated the total BSA for the predicted VHL-PROTAC-SMARCA2 complexes (Fig. 6b). Consistent with the experimental data, our predictions also show a generally negative correlation between total BSA and $ln(K_{LPT})$, supporting the findings of Wurz et al.

In a separate study, ref. 49 investigated the effect of PROTAC linker length on degradation potency, using cereblon (CRBN) as the E3 ligase to induce degradation of Bruton's tyrosine kinase (BTK). They synthesized 11 PROTACs with varying linker lengths (PROTACs 1–11) and found that longer PROTACs (6–11) yield detectable ternary complex formation via fluorescence resonance energy transfer (FRET) and demonstrated potent cellular BTK degradation. In contrast, shorter PROTACs (1–4) showed weak or no FRET signals and were ineffective in cells. PROTAC (5) displayed intermediate behavior.

To further explore the relationship between degradation potency and ternary structure, we used DeepTernary to predict the ternary structures induced by these 11 PROTACs and computed their total BSA. The results, illustrated in Fig. 6c, indicate that as linker length increases, total BSA decreases sharply at first before plateauing. This trend correlates negatively with degradation potency, consistent with the findings from Zorba et al. For the predicted structures of PROTACs (1–4), severe atom clashes between proteins

lead to higher BSAs (left side of Fig. 6d), which explains their inability to form stable ternary complexes and induce degradation. In contrast, for PROTACs (5–11), the increased linker length allows for more flexibility, reducing atomic clashes (right side of Fig. 6d) and facilitating productive protein–protein interactions, which correlate with effective degradation.

Although both Wurz et al. and Zorba et al. demonstrated strong correlations between PROTAC degradation potency and factors like BSA and linker length, it remained unclear whether the observed relationships for VHL-SMARCA2 PROTACs could be generalized to CRBN-BTK PROTACs. By employing DeepTernary to model the ternary structures and calculate BSA for all PROTACs whose degradation potency was experimentally validated, we were able to compare their results and examine these conclusions more thoroughly. In the VHL-SMARCA2 system (Fig. 6b), higher BSA correlates with higher degradation potency (lower $log(K_{LPT})$), whereas in the CRBN-BTK system (Fig. 6c), higher BSA–indicative of shorter linker lengths–is associated with lower degradation potency, highlighting conflicting trends (Supplementary Fig. 4).

By synthesizing the findings from both studies (Fig. 6b, c), we conclude that their conclusions do not inherently conflict. This is because the correlation between BSA and degradation potency appears to be more nuanced than a simple linear correlation. In Fig. 6a, we analyzed the BSA range for 22 known PROTAC-induced complexes with experimentally determined structures, highlighting a range of 1175 to 1422 Å² (shaded gray). When comparing the BSA values of PROTAC-induced complexes in Fig. 6b, c, we found that PROTACs tend

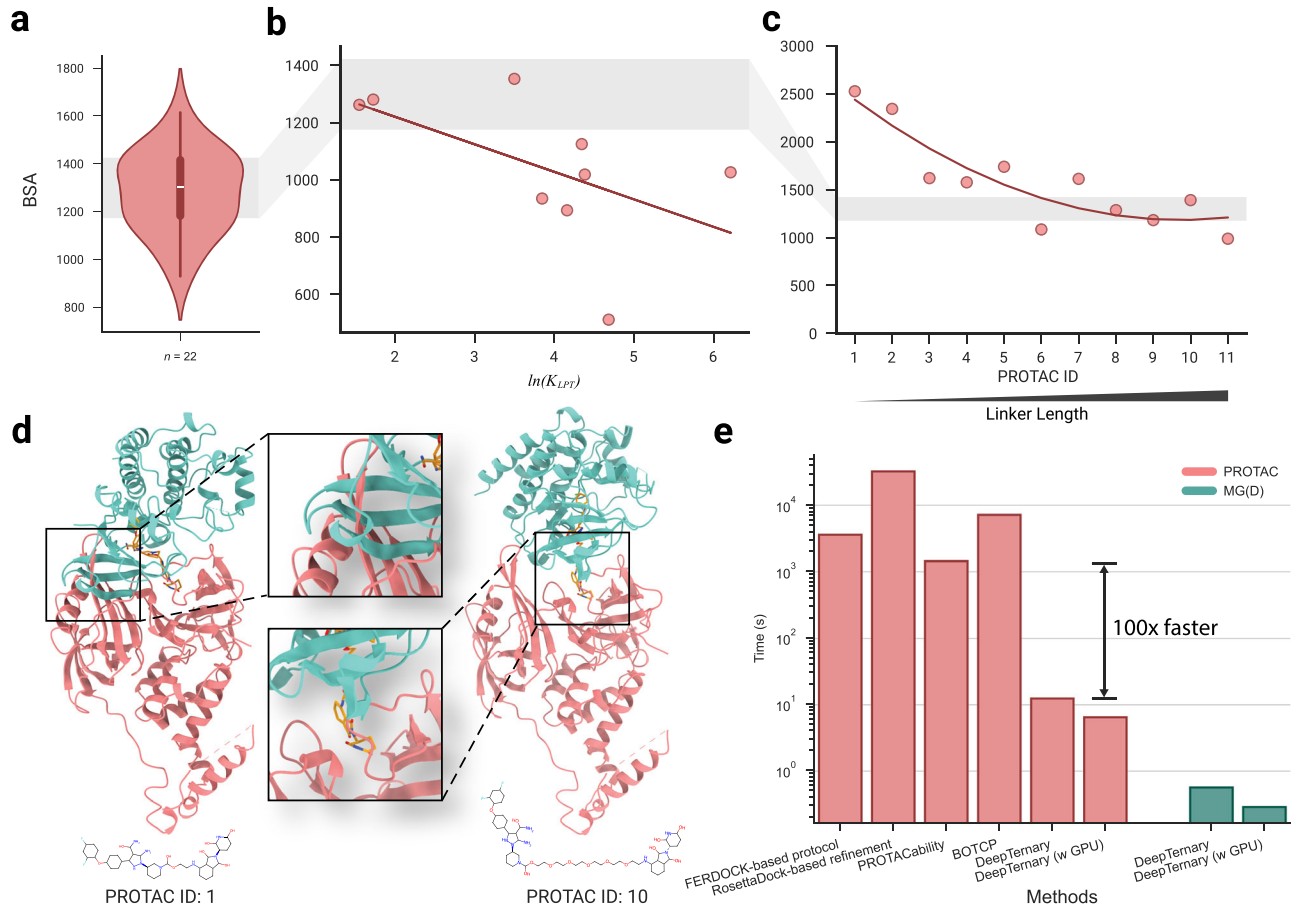

**Fig. 6 | The buried surface area (BSA) based on our predicted structures correlates with degradation potency. a** The BSA of 22 known PROTAC-induced ternary complexes from experimental structures. **b** Correlation analyses between BSA and $ln(K_{LPT})$ for the predicted BRD4-VHL ternary complex with various PROTACs. The quantity $ln(K_{LPT})$ has been proven in previous work[19] to have a positive relationship with $ln(DC_{50})$. The red line represents a linear regression of the points. **c** Correlation analyses between BSA and linker length for predicted CRBN-BTK complexes induced by 11 different PROTACs. As PROTAC ID increases, the linker

length increases. The red line shows a second-order polynomial regression. Notably, PROTACs (6–11) are associated with significant cellular knockdown, while shorter PROTACs(1–4) exhibit weak/no degradation capability. **d** Predicted ternary structures for CRBN-BTK complexes induced by PROTAC (1) and PROTAC (10), illustrating severe atom clashes in PROTAC (1) and increased flexibility to avoid clashes in PROTAC (10). **e** DeepTernary predicts ternary structures ~100 times faster than existing methods.

to exhibit higher degradation potency when their total BSA falls within the 1100 to 1500 Å² range. This suggests that BSA could be a useful metric for virtual screening and inform future PROTAC design.

## DeepTernary is significantly faster than existing methods

Existing methods for predicting ternary structures often require generating numerous candidate structures and applying multiple filtering criteria to identify the most viable options. For instance, Weng et al. utilized a multi-step protocol involving FRODOCK for local docking, followed by energy scoring with Open Babel Obenergy and AutoDock Vina, and further refinement using RosettaDock[24]. This approach, while effective, is time-consuming, taking approximately one hour on an 18-core CPU for the FRODOCK-based process alone, with RosettaDock-based refinement adding another nine hours. More recent methods like BOTCP, which employ Bayesian optimization to expedite candidate sampling, have reduced the process to around 2 h[36].

In contrast, DeepTernary introduces a substantial leap in efficiency by leveraging an end-to-end neural network that embeds learned knowledge directly into its parameters. Unlike traditional docking-based techniques that rely on iterative candidate generation and refinement, DeepTernary predicts PROTAC ternary structures in a fraction of the time. Using 40 seeds, it can predict a ternary

complex in just 12.37 seconds on a 15-core CPU, and as little as 6.48 s with GPU acceleration. For MG(D) complexes, the process is even faster, requiring only a single forward pass of the embedded graphs, yielding results in under 1 second (Fig. 6e). It is worth noting that this time includes both the model's forward time and the data preprocessing time (such as using RDKit to generate initial conformations and file operations), making it instructive for real-world applications. The model-only forward time is reported in Supplementary Table 7.

This dramatic improvement in prediction speed has the potential to revolutionize drug discovery by facilitating the rapid in silico screening of a significantly large number of candidates, making it feasible to explore a broader range of compounds in less time.

## Discussion
In this study, we introduced DeepTernary, a novel deep learning framework consisting of an SE(3)-equivariant graph neural network and a pocket point decoder to predict ternary complex structures induced by PROTACs and MG(D)s. DeepTernary offers a powerful tool for drug discovery by modeling complex interactions within ternary complexes, enabling the optimization of key drug characteristics such as selectivity and potency. Unlike traditional docking methods, which rely on predefined strategies, DeepTernary learns the underlying

physical-chemical rules governing ternary complex formation, resulting in both improved prediction accuracy and significant reductions in computational time. This allows for rapid screening of PROTAC libraries across different E3 ligases and protein targets, providing structure-guided insights for drug development. The model's ability to correlate buried surface area (BSA) with degradation potency further enhances its utility in designing more potent degraders. Additionally, DeepTernary excels in predicting low-affinity, transient interactions for MG(D)s, overcoming limitations of traditional methods and supporting the growing interest in MG(D)s as therapeutics with distinct mechanisms of action.

While DeepTernary is a significant advance, it shares a common limitation with many data-driven approaches: its dependence on large datasets and potential susceptibility to biases present in the training set. Although we have collected a broad dataset from the PDB, there is still room for improvement. Expanding the training data and incorporating lower-resolution experimental datasets could further enhance the model's accuracy and applicability. Additionally, for MG(D)s, we benchmarked using bound structures due to the substantial effort required to obtain unbound structures. Updating the model to predict structure directly from sequence can help address this limitation. We believe that future developments in this direction will further extend DeepTernary's impact and enable broader application in drug discovery.

In conclusion, DeepTernary offers a fast and accurate approach for predicting ternary complexes, representing a valuable tool in the development of TPD therapeutics. In addition, the BSA calculated from generated complexes by DeepTernary may offer valuable insights into the degraded potency, potentially facilitating the structure-guided TPD design. By refining this framework and integrating additional structural data, we anticipate even greater contributions to the field of targeted protein degradation.

## Methods

### Data collection and filtering

In our quest to identify potential ternary complexes, we searched the Protein Data Bank (PDB) to extract potential ternary complexes, applying filters to select structures with at least two proteins and more than one small molecule. This initial filtration process yielded 46,797 potential PDB entries. Subsequently, the filtered candidates were further refined by selecting only X-ray structures that met our high-quality standards-specifically, those with a resolution of 3.5 Å or better and an R-free value of 0.26 or lower-thereby refining our dataset to 22,221 PDB IDs.

From these entries, we extracted 42,441 ternary complexes, some of which included multiple complexes within a single entry, such as assemblies (e.g., 5T35_D_A_759 and 5T35_H_E_759) and instances where different ligands interacted with the same protein pair (e.g., 6ZO8_B_C_LPX and 6ZO8_B_C_PTY). To ensure meaningful protein-small molecule interactions, we imposed additional criteria: the small molecule must share a chain ID with one of the proteins, and the protein components must meet a minimum length requirement of seven amino acids for PROTACs and three for MG(D)s. This was exemplified by the TRAP motif in PDB ID 4TR9[50]. Such stringent criteria effectively pruned nearly half of the initial complexes, leaving us with 25,756 viable candidates for further analysis.

In our final step to validate meaningful protein–ligand interactions, we implemented a two-tiered filtering approach. First, we excluded complexes where the ligand established fewer than three contacts with the protein, defined as ligand atoms positioned within 4 Å of any protein atom. Second, we removed complexes exhibiting steric clashes, identified as any heavy atom pair (one from the ligand and one from the protein) separated by less than 2 Å. While this stringent criteria led to the exclusion of some well-characterized PROTACs and MG(D)s, such as PDB ID 6HAX[51] (R-free = 0.268), which marginally exceeded our 0.26 threshold, and PDB ID 6BN7[52] (chain B ligand clash of 1.97 Å), these structures were manually curated and retained in our database due to their established significance. Additionally, many ligands in the PDB are crystallization buffers that frequently appear across numerous PDB entries and are not functionally relevant. To address this, we manually exclude commonly occurring ligands such as ACT, GOL, PEG, SO4, TRS, XYP, BME, EDO, PG4, and PG5 from the dataset. The culmination of our efforts resulted in a comprehensive structure collection dataset comprising 22,303 complexes.

### Similarity-based dataset splitting

To mitigate the risk of test data leakage and to prevent model overfitting, we adopted a similarity-based dataset splitting strategy. This approach was designed to rigorously evaluate the model's generalization capabilities by ensuring that training complexes were not similar to those in the validation and test sets.

We utilized MMseqs2[40], a highly efficient toolkit for sequence clustering, to group proteins based on a minimum sequence identity threshold of 50%. This involved clustering proteins with similar sequences. Any cluster containing a test set complex was designated as a test cluster. To maintain the integrity of our training set, we excluded all complexes within the test clusters from the training set. Specifically, any protein complex with a sequence similarity exceeding 50% to any test complex was removed from the training data and, if the ligand was dissimilar to those in the test set (Tanimoto similarity <0.85), assigned as a validation sample.

For PROTACs, the 22 known-structure test complexes were clustered into seven groups, resulting in the exclusion of 16 test-similar complexes from the training set, all of which were PROTAC-induced, ensuring the validation set's relevance. 11 of them were chosen as the validation set. For MG(D)s, the 94 test complexes clustered into 44 groups, resulting in 182 excluded training set complexes, of which 137 were chosen as the validation set. This rigorous approach ensured the test set's novelty and provided a robust evaluation of the model's generalization ability.

### Featurization

Following the EquiBind approach[34], both the ligand and proteins were encoded as geometric graphs using the k-nearest neighbor method, and their features were extracted from multiple sources of knowledge, including traditional chemical information and learned amino acid type embeddings[53]. Specifically, in the ligand graph $\mathcal{G}_{lig} = (\mathcal{V}_{lig}, \mathcal{E}_{lig})$, each node (representing an atom) $v_i \in \mathcal{V}_{lig}$ was characterized by atom attributes $\boldsymbol{f}_i$ (a list feature of atomic number, chirality, total degree, formal charge, number of implicit hydrogens, number of hydrogen, radical electrons, hybridization state, aromaticity, and ring participation) and a 3D position vector $\boldsymbol{x}_i \in \mathbb{R}^3$. Edges $\mathcal{E}_{lig}$ were defined between atoms within a distance of less than 4 Å, determined by relative Euclidean distances and bond angles. For the protein graphs $\mathcal{G}_{p1} = (\mathcal{V}_{p1}, \mathcal{E}_{p1})$ and $\mathcal{G}_{p2} = (\mathcal{V}_{p2}, \mathcal{E}_{p2})$, nodes were defined as amino acid type and edges were defined similarly as ligand.

For PROTACs, we utilized the known-pocket unbound evaluation protocol from previous studies[24,27–29,36]. This protocol requires prior knowledge of the unbound structures of both anchor-E3 ligase and warhead-POI binary complexes during inference—a standard practice in PROTAC discovery. To integrate pocket information into Deep-Ternary, we introduced pocket embeddings for graph nodes associated with pockets. These embeddings were integrated into the node features by summation. During training, pocket node coordinates were replaced with their actual values (after random rotation and transformation), while during inference, pocket coordinates from unbound pockets were used. Note that we had ensured that the atom indexes of the unbound pockets and the candidate complexes were well aligned beforehand.

## Model architecture

DeepTernary leverages an SE(3)-equivariant graph neural network along with the attention mechanism, allowing invariant message passing regarding the atom attributes and equivariant message passing regarding the atom coordinates. The model accepts inputs in various formats: the structures of two proteins (E3 ligase and PoI) in PDB or CIF format and the 2D geometry of the small molecule derived from SMILES strings or files in PDB, mol2, or structure-data file (SDF) format. Initially, the RDKit tool was employed to generate possible coordinates of the small molecule. Subsequently, the proteins and the ligand were represented as geometric graphs. The model is fundamentally composed of two primary components: the encoder and the decoder. The encoder learns SE(3)-invariant semantic features and SE(3)-equivariant coordinates, while the decoder outputs pocket points and predicted aligned errors. The comprehensive network architecture of Deep-Ternary was already depicted in Fig. 1c.

**Encoder.** After obtaining the graph representations of the proteins and the ligand, we employed the independent E(3)-equivariant graph matching network (IEGMN)[32] by extending its input from binary complex to ternary complex, in order to facilitate interactions among triplets. This extension involves a series of layers where node coordinates and feature embeddings were updated through both in-graph and cross-graph message passing. Unlike the original IEGMN, our extension allowed for feature updates in a triplet-wise fashion, enabling each monomer to update its features with the awareness of the other two monomers. The update of the coordinates maintains E(3)-equivariance, ensuring that the output faithfully mirrors any independent rotations and translations applied to the input. Formally, there are totally $M$ encoder layers and the latent embedding $\boldsymbol{h}_i^{l+1}$ and node coordinate $\boldsymbol{x}_i^{l+1}$ at the $(l+1)$-th layer were computed as follows:

(1) Intra-graph message passing, which updates edge and node latent embeddings:

$$\boldsymbol{m}_{j\to i} = \phi^e(\boldsymbol{h}_i^{(l)}, \boldsymbol{h}_j^{(l)}, ||\boldsymbol{x}_i^{(l)} - \boldsymbol{x}_j^{(l)}||^2, \boldsymbol{f}_{j\to i}), \forall (i,j) \in \mathcal{E}_{lig} \cup \mathcal{E}_{p1} \cup \mathcal{E}_{p2}, \quad (1)$$

$$\boldsymbol{m}_i = \frac{1}{|\mathcal{N}(i)|} \sum_{j\in\mathcal{N}(i)} \boldsymbol{m}_{j\to i}, \forall i \in \mathcal{V}_{lig} \cup \mathcal{V}_{p1} \cup \mathcal{V}_{p2}, \quad (2)$$

(2) Ternary inter-graph message passing: For the nodes of the ligand, the message from the nodes of the other two graphs $\mathcal{V}_{p1} \cup \mathcal{V}_{p2}$ was computed by

$$a_{j\to i} = \frac{\exp\left(<\phi^q(\boldsymbol{h}_i^{(l)}), \phi^k(\boldsymbol{h}_j^{(l)})>\right)}{\sum_{j'\in\mathcal{V}_{p1}\cup\mathcal{V}_{p2}} \exp\left(<\phi^q(\boldsymbol{h}_i^{(l)}), \phi^k(\boldsymbol{h}_{j'}^{(l)})>\right)}, \forall i \in \mathcal{V}_{lig}, \quad (3)$$

$$\mu_{j\to i} = a_{j\to i}\boldsymbol{W}\boldsymbol{h}_j^{(l)}, \mu_i = \sum_{j\in\mathcal{V}_{p1}\cup\mathcal{V}_{p2}} \mu_{j\to i}, \forall i \in \mathcal{V}_{lig}, \quad (4)$$

We also derived cross-graph message $\mu_i$ for the nodes $\mathcal{V}_{p1}$ and $\mathcal{V}_{p2}$ similar to the above processes.

(3) Calculation of the new node coordinates and embeddings:

$$\boldsymbol{x}_i^{(l+1)} = \Psi\left(\boldsymbol{x}_i^{(l)} + \sum_{j\in\mathcal{N}(i)} \frac{\boldsymbol{x}_i^{(l)} - \boldsymbol{x}_j^{(l)}}{\|\boldsymbol{x}_i^{(l)} - \boldsymbol{x}_j^{(l)}\|} \phi^x(\boldsymbol{m}_{j\to i})\right), \quad (5)$$

$$\boldsymbol{h}_i^{(l+1)} = (1-\beta)\cdot\boldsymbol{h}_i^{(l)} + \beta\cdot\phi^h(\boldsymbol{h}_i^{(l)}, \boldsymbol{m}_i, \mu_i, \boldsymbol{f}_i), \forall i \in \mathcal{V}_{lig} \cup \mathcal{V}_{p1} \cup \mathcal{V}_{p2}. \quad (6)$$

Here, $\phi^e, \phi^x, \phi^h, \phi^q, \phi^k$ denote multi-layer perceptrons (MLPs), $\boldsymbol{f}_{j\to i}$ and $\boldsymbol{f}_i$ represents the initial edge and node features (prior to processing through the IEGMN layers), separately, $\mathcal{N}(i)$ collects all the neighbors

of node $i$, $a_{j\to i}$ indicates the SE(3)-invariant cross-attention coefficient ($a_{i\to i}$ indicates the self-attention coefficient), $<\cdot>$ computes the inner product of two vectors, $\boldsymbol{W}$ is a learnable matrix that transforms latent embeddings according to the cross-attention coefficients, $\Psi$ is a function that imposes distance geometric constraints[34], and $\beta$ is a trade-off parameter. After the encoder process, the latent embeddings and coordinates of all nodes across the three graphs were updated to reflect the intricate interactions among the triple molecules. The predicted coordinates of the ligand ($\boldsymbol{x}_i^l, \forall i \in \mathcal{V}_{lig}$) were assumed to represent the ligand's final conformation within the predicted ternary complex.

Given a ternary complex composed of protein1, a ligand, and protein2 (p1-lig-p2), the predicted structure should be invariant to the order of the proteins. In other words, the predicted structure should be the same regardless of whether the input is (p1-lig-p2) or (p2-lig-p1). To learn this symmetry, the two protein encoders share parameters to learn generalized protein features. During training, the p1 and p2 were randomly swapped for data augmentation.

**Decoder.** For the prediction of ternary structures, we use the ligand conformation derived from IEGMN and require two pairs of pocket points to rigidly align the second protein (protein2) and the ligand with the first protein (protein1), forming a complex. Additionally, the model must predict the predicted alignment error (PAE) for protein2 to assess the quality of the prediction. To this end, we designed a Transformer-based decoder to extract necessary information from graph embeddings. We designed two different decoders for MG(D)s and PROTACs owing to their different MOAs.

Specifically, for MG(D)s, we defined two pairs of pocket points: $(\mathcal{P}_{lig}, \mathcal{P}_{p1\to lig})$ and $(\mathcal{P}_{p2}, \mathcal{P}_{p1\to p2})$. The first pair represents the pocket points between the ligand and protein1, where $\mathcal{P}_{lig}$ denotes the ligand pocket bound to protein1, and $\mathcal{P}_{p1\to lig}$ denotes the protein1 pocket bound to the ligand. Similarly, the second pair represents the pocket points between protein2 and protein1, with $\mathcal{P}_{p2}$ denoting the protein2 pocket bond to protein1, and $\mathcal{P}_{p1\to p2}$ denoting the protein1 pocket bond to protein2. Their corresponding queries are matrices $\boldsymbol{Q}_{lig}, \boldsymbol{Q}_{p1\to lig}, \boldsymbol{Q}_{p2}, \boldsymbol{Q}_{p1\to p2}$, each row of which denotes the query of each node. In addition, we denote the PAE query as $\boldsymbol{q}_{PAE}$. All these values were initialized randomly and processed through an $N$-layer decoder. Each layer requires computing the attention function, represented as Attn($\boldsymbol{Q}, \boldsymbol{K}, \boldsymbol{V}$):

$$\text{Attn}(\boldsymbol{Q}, \boldsymbol{K}, \boldsymbol{V}) = a(\boldsymbol{Q}, \boldsymbol{K})\boldsymbol{W}\boldsymbol{V}, \quad (7)$$

where $\boldsymbol{Q}$, $\boldsymbol{K}$ and $\boldsymbol{V}$ represent the querie, key, and value matrices, respectively; $a(\boldsymbol{Q}, \boldsymbol{K})$ returns the attention matrix, and its element of the $i$th row and $j$th column was given by the attention coefficient $a_{j\to i}$ defined in Eq. (3). When $\boldsymbol{Q}$, $\boldsymbol{K}$ and $\boldsymbol{V}$ become the same, we call it self-attention, otherwise, we call it cross-attention.

We now introduce how to process the queries $\boldsymbol{Q}_{lig}, \boldsymbol{Q}_{p1\to lig}, \boldsymbol{Q}_{p2}, \boldsymbol{Q}_{p1\to p2}$ and $\boldsymbol{q}_{PAE}$, with the information of the hidden embeddings obtained from the encoder before. We first conducted column-wise concatenation:

$$\boldsymbol{Q} = \boldsymbol{Q}_{lig} \| \boldsymbol{Q}_{p1\to lig} \| \boldsymbol{Q}_{p2} \| \boldsymbol{Q}_{p1\to p2} \| \boldsymbol{q}_{PAE}, \quad (8)$$

where $\|$ denotes column-wise concatenation. For conciseness, we collect the updated coordinates and embeddings of the final layer in the encoder over all nodes as $\boldsymbol{X}$ and $\boldsymbol{H}$, henceforth. Specifically for $\boldsymbol{H}$ we further involved the graph embedding features $\boldsymbol{e}$ in order to distinguish the graph identity:

$$\boldsymbol{H} = \left(\|_{i\in\mathcal{V}_{lig}}(\boldsymbol{h}_i + \boldsymbol{e}_{lig})\right) \| \left(\|_{j\in\mathcal{V}_{p1}}(\boldsymbol{h}_j + \boldsymbol{e}_{p1})\right) \| \left(\|_{k\in\mathcal{V}_{p2}}(\boldsymbol{h}_k + \boldsymbol{e}_{p2})\right), \quad (9)$$

Then the pocket queries $Q$ were updated with the following attention layer:

$$Q = \text{Attn}(Q, Q, Q), \tag{10}$$

$$Q' = \phi(\text{Attn}(Q, H, H)), \tag{11}$$

$$H' = \text{Attn}(H, Q', Q'). \tag{12}$$

where $\phi$ is a learnable MLP. We repeated the above attention layer several times. The final queries and embeddings were unfolded as:

$$Q'' = \phi(\text{Attn}(Q', H', H')). \tag{13}$$

$$Q_{\text{lig}}, Q_{\text{p1}\rightarrow\text{lig}}, Q_{\text{p2}}, Q_{\text{p1}\rightarrow\text{p2}}, q_{\text{PAE}} = \text{unfold}(Q''), \tag{14}$$

$$H_{\text{lig}}, H_{\text{p1}}, H_{\text{p2}} = \text{unfold}(H'). \tag{15}$$

For the pocket coordinates, we first computed the attention values between the queries of each local pocket and the embeddings of the corresponding global graph. We then derived the coordinates of each pocket atom as a weighted sum of the coordinates of the entire graph. Specifically, we computed:

$$P_{\text{lig}} = a(Q_{\text{lig}}, H_{\text{lig}})X_{\text{lig}}, \tag{16}$$

$$P_{\text{p1}\rightarrow\text{lig}} = a(Q_{\text{p1}\rightarrow\text{lig}}, H_{\text{p1}})X_{\text{p1}}, \tag{17}$$

$$P_{\text{p2}} = a(Q_{\text{p2}}, H_{\text{p2}})X_{\text{p2}}, \tag{18}$$

$$P_{\text{p1}\rightarrow\text{p2}} = a(Q_{\text{p1}\rightarrow\text{p2}}, H_{\text{p1}})X_{\text{p1}}, \tag{19}$$

where the matrices $P$ denote the predicted pocket point coordinates.

The PAE $q_{\text{PAE}}$ was estimated using an MLP, reflecting the prediction confidence. Given the computational intensity of real-time DockQ score calculations, we use the root mean square deviation (RMSD) between predicted and actual coordinates of protein2 as a training surrogate for PAE. With predicted pocket points and the ligand conformation, the final ternary complex structure is assembled, which will be detailed in the next subsection.

For PROTACs, we directly bound the two proteins at their two ends, and designed two protein–ligand pocket coordinates: $(P_{\text{p1}}, P_{\text{lig}\rightarrow\text{p1}})$ and $(P_{\text{p2}}, P_{\text{lig}\rightarrow\text{p2}})$, representing pockets of (E3, anchor) and (POI, warhead), respectively. Different from MG(D)s, these pocket points are already known from the unbound structures. Thus, without the need of the computations above derived for MG(D)s, $P_{\text{p1}}$ and $P_{\text{p2}}$ were directly taken from the unbound protein structures, $P_{\text{lig}\rightarrow\text{p1}}$ and $P_{\text{lig}\rightarrow\text{p2}}$ were taken from the predicted ligand coordinates $X$ from the encoder according to unbounded pocket masks. The decoder predicts the PAE for PROTACs using the same architecture but with only the PAE query reserved. In other words, we conducted Equation (10) - Equation (13) by setting $Q = q_{\text{PAE}}$.

### Transformation to generate the final output

Considering the different modes of action of PROTAC and MGD, we adopted two slightly different ways to construct the final complex structure. PROTAC molecules comprise three elements: the anchor, warhead, and connecting linker. The anchor and warhead are typically selected from known bounded ligands to E3 ligase and the POI, respectively. This selection facilitates rational design, leveraging existing unbound binding data between the anchor and E3, as well as the warhead and PoI, to construct the complex structure. Following this process,

the PROTAC was first aligned with the unbound pocket of E3 (protein1) based on the predicted pocket points for the anchor. The linker and warhead coordinates were determined according to the conformation of the PROTAC. Subsequently, the coordinates of POI (protein2) were determined by aligning its unbound structures to the aligned warhead positions according to predicted protein2 pocket points:

$$R_{\text{lig}}, t_{\text{lig}} = \text{kabsch}(P_{\text{lig}\rightarrow\text{p1}}, P_{\text{p1}}), \tag{20}$$

$$x_{\text{lig}} = (R_{\text{lig}}x_{\text{lig}}^\top)^\top + t_{\text{lig}}, \tag{21}$$

$$P'_{\text{lig}\rightarrow\text{p2}} = (RP_{\text{lig}\rightarrow\text{p2}}^\top)^\top + t, \tag{22}$$

$$R_{\text{p2}}, t_{\text{p2}} = \text{kabsch}(P_{\text{p2}}, P'_{\text{lig}\rightarrow\text{p2}}), \tag{23}$$

$$x_{\text{p2}} = (R_{\text{p2}}x_{\text{p2}}^\top)^\top + t_{\text{p2}}, \tag{24}$$

where kabsch denotes the Kabsch algorithm[54], $^\top$ denotes matrix transpose.

A more direct alignment approach was employed for MG(D)s. Both the ligand and protein2 were aligned directly to protein1. This was achieved by predicting the pocket points of interaction between protein1 and the ligand, as well as between protein1 and protein2. The decoder's predicted pocket points facilitated the alignment of the ligand and protein2 to protein1, resulting in the final ternary complex structure:

$$R_{\text{lig}}, t_{\text{lig}} = \text{kabsch}(P_{\text{lig}}, P_{\text{p1}\rightarrow\text{lig}}), \tag{25}$$

$$x_{\text{lig}} = (R_{\text{lig}}x_{\text{lig}}^\top)^\top + t_{\text{lig}}, \tag{26}$$

$$R_{\text{p2}}, t_{\text{p2}} = \text{kabsch}(P_{\text{p2}}, P_{\text{p1}\rightarrow\text{p2}}), \tag{27}$$

$$x_{\text{p2}} = (R_{\text{p2}}x_{\text{p2}}^\top)^\top + t_{\text{p2}}. \tag{28}$$

### Training and inference

During the training process, protein structures were derived from bound structures and ligand conformations generated by the RDKit toolkit[41]. For each training ligand, we pre-generated a pool of 50 random conformations. In each training iteration, protein1 or protein2 was randomly fixed, while the other protein and a randomly selected conformation from the ligand's 50-conformation pool were subjected to random rotations and translations from their original positions. Coordinates were normalized before being input into the model to stabilize the training process, with random noise added to graph features and coordinates to avoid overfitting.

The model was trained with six losses to guide it toward generating accurate outputs. The total loss is formulated as follows:

$$\mathcal{L} = \mathcal{L}_{\text{lig}} + \mathcal{L}_{\text{kabsch\_lig}} + \mathcal{L}_{\text{ot1}} + \mathcal{L}_{\text{ot2}} + \mathcal{L}_{\text{intersection}} + \mathcal{L}_{\text{PAE}}. \tag{29}$$

where $\mathcal{L}_{\text{lig}}$ indicated the mean squared error (MSE) loss between the predicted and ground-truth ligand coordinates, and $\mathcal{L}_{\text{kabsch\_lig}}$ denoted the MSE loss after rigid alignment of the predicted ligand to the ground truth using the Kabsch algorithm[54]. $\mathcal{L}_{\text{ot1}}$ and $\mathcal{L}_{\text{ot2}}$ corresponded to the optimal transport loss[55] between the predicted pocket points and target pocket coordinates. $\mathcal{L}_{\text{intersection}}$ represented the intersection punishment between proteins and the ligand, and $\mathcal{L}_{\text{PAE}}$ indicated the predicted aligned error of protein2, calculated using the L1 loss between the predicted and ground-truth RMSD of protein2.

During inference, unbound structures were used for PROTACs and bound structures for MG(D)s. The initial ligand conformations were randomly generated by RDKit using different seeds. For each PROTAC, we performed 40 samplings and ranked the results based on predicted PAEs. For molecule glues, only one sampling was performed due to their limited atom numbers and conformational flexibility.

DeepTernary contains 16.73 million parameters and was trained for about 8 h on two Nvidia L40 GPUs.

## Calculation of buried surface area (BSA)

The buried surface area (BSA) was calculated using ChimeraX[56]. Ligands were assigned unique sequence IDs separate from proteins, and the "interfaces" command computed solvent-accessible surface area (SASA) for each interacting chain pair within the complexes. The total BSA was determined by summing the SASA values across all protein–protein and protein–ligand interactions. The BSA represented in Fig. 6 is the average BSA of the top five most confident (lower PAE) predictions from DeepTernary.

## Evaluation metrics

Following recent studies[24,36], we adopted the **DockQ score**[57] as a quantitative measure to evaluate prediction quality. The DockQ score is a continuous metric ranging from 0 to 1, calculated based on three components: $F_{nat}$, LRMS, and iRMS. $F_{nat}$ represents the fraction of native contacts maintained in the predicted complexes. LRMS is the root mean square deviation (RMSD) between backbone atoms after aligning the predicted structure to the native one. iRMS is the RMSD of backbone atoms of the interface residues. By integrating these three criteria, the DockQ score provides a comprehensive measure of prediction quality, with higher values indicating higher-quality predictions.

To compare our methods with previously published approaches, we also calculated the fraction of acceptable predictions and compared them with other methods. It is worth noting that the criteria for an "acceptable" prediction vary across different studies. We categorize these criteria as follows:

**DockQ >0.23.** This threshold indicates a quality prediction based on the DockQ scoring system.

**CAPRI criterion.** Derived from the critical assessment of predicted interaction (CAPRI)[58], predictions are classified into high, medium, or acceptable. This criterion has been employed to assess the quality of PROTAC-induced complex predictions, as used by ref. 29.

**RMSD <10** Å. This criterion involves calculating the Cα RMSD and is commonly used as the upper limit for an "acceptable" pose in protein–protein docking contexts. It is straightforward and easy to compute.

For PROTACs, given the model's generation of multiple predictions from varying initial conformations, we employ the **Acceptable Rank** metric, following existing methods. This metric is determined by sorting predictions based on their predicted alignment error (PAE) and identifying the rank of the first prediction achieving a DockQ score greater than 0.23.

By applying these metrics, we ensure a robust evaluation of our model's performance in predicting ternary complex structures.

## Reporting summary

Further information on research design is available in the Nature Portfolio Reporting Summary linked to this article.

## Data availability

The TernaryDB complex list, cluster results of the training dataset, training log, pretrained model weights, and unbound structures for PROTAC testing have been deposited at https://github.com/youqingxiaozhua/DeepTernary for public access. A snapshot of the TernaryDB dataset is stored in Zenodo at https://doi.org/10.5281/zenodo.15514874. The complex crystal structures are downloaded from the Protein Data Bank at https://www.rcsb.org/. Source Data are provided with this paper. The PDB IDs analysed in this manuscript are: 5T35, 6W7O, 6W8I, 7BQU, 4JDD, 4TR9, 6HAX, and 6BN7. Source data are provided with this paper.

## Code availability

Codes for running DeepTernary have been released on GitHub and are free for academic, personal, and commercial use at https://github.com/youqingxiaozhua/DeepTernary. A snapshot of the current version is stored in Zenodo at https://doi.org/10.5281/zenodo.15515197[59].

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

## Acknowledgements

The work of Y.Y. was in part supported by the National Science and Technology Major Project (2023ZD0120801). W.H. was jointly supported by the National Natural Science Foundation of China (No. 62376276), Beijing Nova Program (No. 20230484278), Beijing Outstanding Young Scientist Program (No. BJJWZYJH012019100020098), the Fundamental Research Funds for the Central Universities, and the Research Funds of Renmin University of China (23XNKJ19). W.D. was supported by the Whitcome fellowship and the Molecular Biology Interdepartmental graduate program at UCLA. J.A.W. was supported by National Institutes of Health grants R01 GM089778 and GM112763 and the David Geffen School of Medicine at UCLA. Computational facilities were provided by the UTS Interactive High-Performance Computer Cluster.

## Author contributions

F.X., W.H. and W.D. conceived the study. F.X. wrote the code and trained the model. F.X. and M.Z. collected and processed training data. F.X., M.Z., S.L., X.G., J.W., W.H., Y.Y. and W.D. contributed to the analysis of the results. F.X., M.Z., W.H. and W.D. wrote the manuscript. W.H., Y.Y. and W.D. offered supervision throughout the project. All authors revised the manuscript.

## Competing interests

The authors declare no competing interests.

## Additional information

Wenbing Huang, Yi Yang or Weixian Deng.

