## [Transparent Peer Review file · Nature Communications]

SE(3)-Equivariant Ternary Complex Prediction Towards Target Protein Degradation

Corresponding Author: Dr Weixian Deng

Version 0:

Reviewer comments:

Reviewer #1

(Remarks to the Author)

This paper tackles a highly interesting problem of ternary structure prediction, which may have profound implications in the design of PROTACS and MGDs, providing a novel targeted protein degradation mechanism for drug discovery. The authors curate a comprehensive dataset and develop a performant SE(3)-equivariant model for such predictive tasks.

Strengths:

- The problem is very well-motivated.
- The writing is clear and organized.
- The graphic illustrations are particularly telling.
- The proposed model seems to be highly performant in predicting the ternary structures.

Weaknesses:

- There is little innovation in terms of the architecture itself. It is a direct application of mature techniques.

Questions:

- In Figure 4e, Could you comment on the higher performance variance of your model?
- In Figure 5e, Why is there little difference for your model with and without GPU?

(Remarks on code availability)

Reviewer #2

(Remarks to the Author)

(Remarks on code availability)

Reviewer #3

(Remarks to the Author)

Xue et al. describe a novel ternary complex prediction deep learning method trained on thousands of non-degrader ternary structures in the PDB that predicts degrader ternary complex structures faster and more accurately than previous methods. The model uses SE(3)-equivariant encoder to distill information in feature and coordinate vectors. This information is decoded by a pocket points decoder to identify where the degrader binds. These points are then used to position the degrader and the second protein. While the study is very novel and highlights the potential for machine learning application in this field, a number of major points should be addressed or clarified before publication.

Major points:

- The last section before the discussion describes the "Optimization of model designs", where they refine the choice and hyperparameters of the final model. Surprisingly, it seems like, while they are refined using the training set, the final selection of these hyperparameters is based on evaluations conducted on the test set. If this is the case, this is a flaw: test set and validation set should not overlap. If this is not the case, it should be clarified. It would be best not to keep this critical information at the very end of the Results section.
- The authors should explain why a pocket points decoder is needed to superimpose the PROTAC back onto the protein when the binding pose of the unlinked ligands are already available. In those cases, the pocket points are known, and the model only adjusts the conformation of the PROTAC in the encoder to fit into the complex. The decoder might be more relevant for molecular glues complexes if the binding site on the target protein is not known. However, the performance for this is poor.
- Many (most?) of the compounds in the training sets are probably molecules from the crystallization buffer and are not functionally relevant. This point should be evaluated and discussed.
- Some of the data are not properly presented in figures. Some data seem to be missing in Figure 3A,B,C where the bottom plots only show 13 data points or less for 22 examples and it is unclear whether data from old benchmark or new benchmark are used (22 vs 43 complexes). Also, in the supplementary data, there a big difference between the first and second benchmark set where 6BOY initially gets a score of 0.7 and then 0.29: these differences are not addressed in the text and it is unclear which data are plotted. This questions the reproducibility of the model and whether the author initially trained the model with the complexes that were later added to the second benchmark dataset. I would recommend the authors to run a reproducibility study to evaluate what the model's performance is over n of 3 or 5. Furthermore, 182 complexes are filtered out based on similarity to MGD complexes but are not added to the benchmarking dataset similar to the PROTAC dataset. This would increase the test set and would have allowed for a validation set. Why was this not added?
- There is a focus on the protein-protein complex performance scores, but not on the PROTAC or MGD. When looking at the figures for the predicted complexes, the PROTACs and MGD are predicted far from the pose in the crystal structure. The authors should provide data on the accuracy of the predicted PROTAC and MGD binding poses. This is only a suggestion, but for PROTACs, the authors may consider keeping the bound ligand(s) rigid during conformational searches and only keep the linker atoms flexible. This would improve the superimposition of the PROTAC back into the proteins. Currently based on the accuracy of the prediction of the compounds DeepTernary seems more a binary complex prediction than a ternary complex prediction.
- The authors should comment on why PROTACs perform better than MGD (DockQ score average of 0.70 and 0.10 respectively) while the training set is skewed towards small molecule complexes.
- I would recommend the authors to look at other small molecule complex prediction software such as Chai-2 and AF-3 to benchmark their model for MGD prediction.
- It is said that the training set is made available, but it is hard to find on the github page. I would highly recommend for the authors to provide at least a list of PDB codes + chains in the supplementary data as csv file for the training set, so other researchers can use the training set for model development. Since, in the second benchmark, several ternary complexes are missing (8BB4, 8BEB, 8DSO, 8FY0, 8FY1, 8FY2, 8PC2, 8QJR, 8QVU, 8QW6, 8QW7, 8R5H) this questions whether these might have ended up in the training dataset causing data leakage.

Minor points:

- Since BSA is highlighted late in the paper, it would be nice to add analysis of BSA of the training set.
- Line 370: The text mentions only 22 complexes for the test set but then in methodology this is increased to 43 complexes, I think the text here should reflect that.
- Line 406: "The acceptable prediction appeared within the top-ranked results" what is considered the top-ranked? Top 10, top 5, top 3? Generally, the sentence "Importantly, the majority of acceptable predictions appeared within the top-ranked results, with approximately half deemed reasonable" is vague, what is considered acceptable and reasonable?
- How different are the PAE scores between the most accurate prediction and other predictions? Are these small difference or big difference? It would be nice to see that PAE score of the most accurate prediction can indeed be used as a confidence score and a filter. I would recommend including a plot to show the correlation between the DockQ scores and the PAE scores to validate that PAE can be used as a confidence score for the predicted complex.
- Line 403: Rank of what? The closest prediction to the crystal structure? How far off was the number one prediction to the crystal structure? How different were the predictions among each other for one complex was there convergence to one ternary complex conformation if sampled enough PROTAC conformations?
- Line 404: Ligands are conformationally sampled, was this done for the training step as well?

- Line 483: "The PAE matrix in the top right corner further highlights AF3's lower confidence": it would be nice to define somewhere in the text what is considered a confident PAE score.
- Line 527: "This suggests that the large, well-folded domains in Group 1 complexes involve more complex binding rules, which may not be adequately covered by the training set": it would be nice if this analysis was performed on the training set to further support this conclusion. It may well be the other way around, where the PDB has more ternary complexes with folded regions and molecules than small recognition motifs.
- Line 538: "They also showed that lower KLPT corresponds to higher ternary complex binding affinities and, consequently, higher degradation potency". Do the authors mean "lower BSA" instead?
- Line 540: How many PROTACs were in this dataset? How strong is the correlation? What is the R^2 ?
- Line 570: The authors conclude that the differing trends likely stem from intrinsic differences between E3, but as reported early, the short linker caused clashes between the proteins. What is the conclusion if you only look at predictions where no clashes are found? If you zoom in on PROTAC 5-11 and plot the correlation between BSA and $\ln(DC50)$ would you see a similar correlation/trend to the Wurz et al paper? It would be good to specify whether the authors recommend for future groups to use the BSA metric as a virtual screening tool and explain why or why not.
- Does the model take into consideration induce fitting of the complexes as observed in PDB: 6HAX.
- I would recommend using swarm plots for all boxplots to show how many test cases there are.

Figure 1:

- B) Disassemble implies that the parts from ternary complex were used for training rather than matched to the monomers of the respective proteins which has been done for the PROTAC complexes. So, would recommend including this in the figure to avoid confusion.
- C) Is the protein missing for the purple stars?
- C) The step where conformations are generated for the PROTAC/MGD is missing in the figure

Figure 2:

- E) the text in the figure legend does not match the figure, no colouring for clusters are visible.
- o "e, Proteome-wide view of the collected dataset. ESM-1b [38] sequence embeddings for the two proteins in each complex are calculated and concatenated. This is followed by two-dimensional (2D) Uniform Manifold Approximation and Projection (UMAP) and Density-Based Spatial Clustering of Applications with Noise (DBSCAN), with clusters distinctly marked using a qualitative color scale. Similar complexes to PROTACs- and MGD(s)-involved ternary structures are denoted as red and green square points, respectively. "

Figure 3:

- Are these boxplots scores for the best predicted complexes for each of the complexes or average for each test case?
- F) In the crystal structure PDB: 6W8I, there are three different poses for the complex, which one was used for comparison?
- F) the last panel shows that the PROTAC does not overlap with the cIAP ligand.
- D) what causes the big difference in DockQ score between BIRC2 and CRBN?
- The scores in the supplementary for the CRBN complexes are 0.70 and 0.71 DockQ, but the average here seems to be 0.50 which does not reflect the data. In supplementary table 2 the results are 0.62 and 0.29 which would make sense to result in 0.50. What caused the difference in result in supplementary table 1 vs supplementary table 2 for the same complex, for example 6BOY?

Figure 4:

- I would recommend also using the high/medium/acceptable, RMSD and rank metrics for the molecular glue dataset. Along with MGD RMSD. The ternary complexes show quite a difference in RMSD in the figure specifically the MGD RMSD which would not be suitable for structure activity optimization if $>1.5\text{\AA}$
- B) include color label for crystal structure

Figure 5:

- B) What is the correlation R^2 ?
- C) How well does BSA reflect the DC50 values?

(Remarks on code availability)

- The github repository could be better organized. I would recommend putting all the code in one folder called DeepTernary. I would import libraries like DockQ and equibind and list them as dependencies for the program unless the code is modified in these libraries.

Reviewer #4

(Remarks to the Author)

SE(3)-Equivariant Ternary Complex Prediction Towards Target Protein Degradation

In this work, the authors propose a deep learning model that leverages SE(3)-equivariant graph neural networks (GNN) to predict the ternary protein complexes that are involved in target protein degradation. The proposed model is evaluated on the MGD benchmark dataset, which is a dataset of 22 known PROTAC-induced ternary structures proposed by Rao et al. in "Bayesian optimization for ternary complex prediction (BOTCP)". The authors compared their method to existing methods, such as FRODock, RosettaDock, and BOTCP. The results show that the SE(3)-equivariant model outperforms the baseline methods in predicting ternary complexes, concluding that SE(3)-equivariant GNNs can be a superior and fast alternative to docking methods.

The authors take great care in avoiding any data leakage between the training and test data. They also provide a detailed description of the model architecture and the loss functions used for training. Nevertheless, the authors could improve the paper by refactoring the order of the sections and the figures reporting their results, as well as adding documentation to the provided code. It currently appears that it would be very hard to reproduce the results of the work, so adding detailed guides on how to re-run the code and reproduce the results would be beneficial.

For reference: my background is mainly in computer engineering and machine learning, with a focus on deep learning and its applications in cheminformatics.

Major Points

- As mentioned above, despite the authors providing a code repository (which is great), there is no clear guide on how to obtain the training data nor on how to actually train the model, should one want to reproduce the results. Also, it is very hard to navigate the directory tree of the project, as it lacks valuable READMEs on how to navigate the codebase.
- It is unclear what are the input and output data for the model. As far as I understand, the encoder part already expects the docked structure of the two proteins, and then outputs the refined 3D coordinates of the ternary complex. How are the pocket points defined and fed to the decoder part? Are the "pocket queries" suggested by the user? The authors briefly attempt to answer these questions in lines 358-361, but I suggest to provide a more detailed explanation and to refer to later sections of the paper for more details. This could also be better clarified in the figures.
- How are the "groups of pocket points" defined? I believe the authors give an explanation in the "Model architecture" section, but it is not clear to me. I suggest to provide a more detailed explanation (or pointers) already when mentioning Figure 1, i.e., in the "Introduction" section.
- The authors are careful to exclude any PROTAC and molecular glue molecules from the training data. This means that the model is trained on solely out-of-distribution data only. Because of that, how can the model generalize to unseen PROTACs and molecular glues? Since the model seems to be able to predict the ternary complex with high scores, how do the train and test data distribution look like? How similar are they? How different?
- Line 357 mentions that the model parameters for the two proteins are shared in order to capture the symmetry of the protein complex. How is this beneficial? In case of PROTACs, POIs and E3 ligases can be quite different, so there would not be much "symmetry information" for the model to leverage. On the contrary, I would expect the model to learn different parameters for the two proteins. Could you provide more details on this choice? I do acknowledge that the model might be faster because of this design choice, but I would still expect a more detailed motivation from the authors.
- The evaluation metric referred as "High/Medium/Acceptable" is not clearly defined. I suggest to provide a more detailed explanation of this metric. I would suggest to add references to the "Evaluation metrics" section and provide a more detailed explanation of the "High/Medium/Acceptable" metric there maybe.
- The IEGMN architecture (which the authors cleverly customized for this work) is well detailed in the "Model architecture" section. However, it is already mentioned before in the "Optimization of model designs" section, without any reference nor pointer to later sections, which is a bit confusing. I suggest to re-order the sections as to first describe the model architecture and then discuss the optimization of the model design, and finally how it performs in terms of computation time.
- Section "Data collection and filtering" under "Methods" is not detailed enough for reproducing and obtaining the training data from the raw PDB structures. Also, it doesn't add significant more details than what already explained in the "Results" section. I suggest to provide a more detailed explanation of the data collection process and eventually pointers to the code repository.
- At lines 377, "To mitigate data leakage, we excluded any similar protein pairs from the dataset used to train our model.": what's the definition of "similar protein pairs" here? What's the threshold used to define similarity? How was this threshold chosen?
- Model is predicting a predicted aligned error (PAE) score. What's the numerical definition of PAE? How are the model predicted PAE values evaluated? Related to the PAE definition, at lines 403-404, what does the sentence "With an average rank of 2.81 under 40 seeds" and Figure 3a middle mean?

Minor Points

- The line system of the paper get mis-aligned, but at roughly Lines 843 and 851, why is the PAE query denoted as "qVAE"? Shouldn't it be "qPAE"?
- Some figures, e.g., Figures 3a, 3b, 3c, 6a, show some asterisks and notes, but it is not clear what they represent or what they point to. I'd suggest to maybe add a legend to the figure to clarify the meaning of these symbols.
- Figure 6a: what's the x axis? Why are data displayed in that way and not via, e.g., a whisker plot?
- Figures 3a, 3b, 3c, and 3d can be greatly improved:
- The color scheme of the comparing methods is not consistent across the figures, nor it's their order.
- Why aren't the comparing methods included in Figures 3d?

- Figure 3b and 3c report a "Method 4B", but I couldn't find a clear reference to what it stands for.
- I would suggest a more contrasting color palette for Figures 3f, 4c, 4d, and 4b.
- Reference 32 is showing some "???" in its Bibliography entry.
- The axis on Figure 6e is referred as "Seed Number", but it should be more like "Number of Sampled Random Conformations".
- There is a typo in the Kabsch loss at line 956.

(Remarks on code availability)

Version 1:

Reviewer comments:

Reviewer #1

(Remarks to the Author)

The revised paper has addressed all of my concerns previously. I therefore recommend acceptance.

(Remarks on code availability)

The revised paper has addressed all of my concerns previously. I therefore recommend acceptance.

Reviewer #2

(Remarks to the Author)

(Remarks on code availability)

The code appears to be a usable resource with sufficient documentation for the users who want to reproduce the results from the paper. The README provides comprehensive instructions for installation and execution, and the availability of pre-trained models and datasets enhances reproducibility. However, there is a issue when I tried to access the TernaryDB dataset. The url link provided in the README file returns "Not Found". I further looked for the test.csv and val.csv in the data folder, but they are not there. This data availability issue might potentially make the result validation difficult.

Reviewer #3

(Remarks to the Author)

I thank the authors for a detailed evaluation of the comments, questions and suggestions. I observed significant improvements to the manuscript, but still encounter one major flaw that has not been properly addressed and one minor flaw that should be discussed. These points should be addressed before publication.

Major:

While I appreciate that the authors identified a validation set, the use of the validation set is not following best machine learning practices. Here are a few papers on the use of validation set and reason why not to use test set for hyperparameter optimization: Walsh et al. 2021 (<https://www.nature.com/articles/s41592-021-01205-4>) (Nature Methods), Chicco and Jurman 2022 (<https://doi.org/10.3389/fdata.2022.979465>), and Lones 2024 (<https://arxiv.org/pdf/2108.02497v5>) (specifically the section "Do avoid sequential overfitting"). In particular, the guide by Lones is helpful for academics to avoid common pitfalls.

Figure 1 should not show results on the test set, but rather the results on the validation set and the paragraphs "The architecture of DeepTernary" and "Effectiveness of model designs" should be written about the validation set instead of the test set. The authors noted that the model performing best on the validation set had similar hyperparameters to the model used to evaluate the test set. I would like to emphasize that this is the reverse of what is normally done in the field and that common practice should be adopted in future studies.

More concerning, the validation set was flagged as being similar to the test set. Therefore, there is data leakage from validation set to test set. Hence, the final results are not a true representation of how the model would generalize to novel PROTAC or MG cases.

Minor:

In Q23 response to reviewers, the authors indicated that they re-docked monomer structures extracted from crystallized ternary complexes. This does not represent real-life case studies where the structure of monomers are not biased towards formation of the complex (e.g. optimal side-chains or loop conformations for protein-protein binding). Therefore, DeepTernary shows good performance in predicting complexes when starting from crystallized ternary complexes, but generalization to prospective cases where a crystal structure is not available is unclear. This should be addressed by testing

DeepTernary with monomeric protein structures, or this limitation should be highlighted and discussed in the manuscript.

Lastly, in the spirit of academic congeniality, the authors may want to try the RDKit code below to keep both chemical handles rigid during conformational searches of a PROTAC:

```
import re
from rdkit import Chem
from rdkit import DistanceGeometry
from rdkit.Chem import rdDistGeom

def adjust_bounds(bounds_mat,map,mol):
    for i in range(len(map)):
        a1=map[i]
        for j in range(i+1,len(map)):
            a2=map[j]
            d = math.sqrt((mol.GetConformer().GetAtomPosition(a1)-mol.GetConformer().GetAtomPosition(a2)).LengthSq()/1)
            if (a1<a2):
                bounds_mat[a1,a2]=d+0.1
                bounds_mat[a2,a1]=d-0.1
            check = DistanceGeometry.DoTriangleSmoothing(bounds)
            return bounds_mat, check

chemical_handles = []
PDBs = ['4TZ4.pdb','3MXF.pdb']
for pdb_code in PDBs:
    handle = ""
    with open(pdb_code, 'r') as f:
        for line in f:
            if re.search('HETATM', line)!=None:
                handle = handle + line
    chemical_handles.append(Chem.MolFromPDBBlock(handle))

suppl = Chem.SDMolSupplier('RN6.sdf')
PROTAC = suppl[0]
PROTACH = Chem.AddHs(PROTAC)
CH1_MAP = PROTACH.GetSubstructMatch(chemical_handles[0])
CH2_MAP = PROTACH.GetSubstructMatch(chemical_handles[1])
bounds = rdDistGeom.GetMoleculeBoundsMatrix(PROTACH)

bounds, check = adjust_bounds(bounds,CH1_MAP,PROTACH)
print(check)
bounds, check = adjust_bounds(bounds,CH2_MAP,PROTACH)
print(check)

ps = rdDistGeom.ETKDGv3()
ps.SetBoundsMat(bounds)
cids = rdDistGeom.EmbedMultipleConfs(PROTACH,5,ps)
writer = Chem.SDWriter("PROTAC_conf.sdf")
for i in range(PROTACH.GetNumConformers()):
    mol_ID = "example" + f"_{i}"
    PROTACH.SetProp('ID', mol_ID)
    writer.write(PROTACH, confId=i)
```

(Remarks on code availability)
I verified that the code is available.

Reviewer #4

(Remarks to the Author)
Thank you to the authors for the thorough revision of the manuscript. Happy to recommend it for publication.

(Remarks on code availability)

Version 2:

Reviewer comments:

Reviewer #3

(Remarks to the Author)

The authors addressed all my comments. In my opinion, this manuscript deserves publication in this Journal.

(Remarks on code availability)

Responds to Reviewer #1

Q1: This paper tackles a highly interesting problem of ternary structure prediction, which may have profound implications in the design of PROTACS and MG(D)s, providing a novel targeted protein degradation mechanism for drug discovery. The authors curate a comprehensive dataset and develop a performant SE(3)-equivariant model for such predictive tasks.

Strengths:

- The problem is very well-motivated.
- The writing is clear and organized.
- The graphic illustrations are particularly telling.
- The proposed model seems to be highly performant in predicting the ternary structures.

Response: We sincerely appreciate your encouraging remarks and valuable insights regarding the merit of our approach and the significance of our work.

Q2: There is little innovation in terms of the architecture itself. It is a direct application of mature techniques.

Response: Thank you for your valuable feedback on the architecture. We appreciate your observation that the main components used in our architecture are based on mature techniques. However, we believe that the key innovation lies in how these components are integrated to effectively address the challenge of ternary structure prediction.

Particularly, as shown in Fig. 1c, the encoder employs a novel ternary inter-graph attention mechanism to capture the geometric relationships among the three input structures. Meanwhile, the decoder utilizes a query-based Transformer, which takes randomly initialized pocket points and predicted alignment error (PAE) tokens as input queries, generating corresponding outputs to construct the final complex structure along with the PAE estimation. These upgrades enable our proposed DeepTernary to generalize to unseen PROTAC and MG(D) structures, which is very important for drug screens. We have further highlighted the above points in the revised version.

Q3: In Figure 4e, could you comment on the higher performance variance of your model?

Response: Thank you for the question. The higher performance variance observed in our model compared to EquiDock can be attributed to the inherent difficulty of the task. For MG(D)s, we performed blind docking—docking without prior knowledge of the pocket information—which is particularly challenging in 3D space due to the high degree of freedom. In this setting, EquiDock consistently yields poor performance, thus its variance remains low. As shown in the figure below, EquiDock failed to predict almost all test samples (with DockQ score close to 0), whereas our DeepTernary successfully predicted some of them with high DockQ (up to 0.6 – 0.8). The MG(D) molecules are of high diversity, as illustrated in Figure 2e and 2f. Some test complexes may share similar interaction mechanisms with the training set, allowing our model to generalize, while others differ significantly, leading to prediction failures. This variation could contribute to the higher performance variance observed in the MG(D) benchmark. We have also added this analysis to the manuscript.

Q4: In Figure 5e, why is there little difference for your model with and without GPU?

Response: Thank you for your question. The reported numbers in Figure 5e represent the total execution time, including data preprocessing (RDKit conformation generation and file operations) as well as model inference. While GPUs are well-suited for accelerating model inference, this step constitutes only a small portion of the overall execution time. As a result, the difference in total runtime with and without a GPU is minimal.

To clarify this aspect, we have now included a direct comparison of model inference time with and without GPU in the Supplementary Materials and have revised the manuscript accordingly.

Revision:

Supplementary Table A6 DeepTernary Forward Pass Time. The execution times reported in the main manuscript include both the model's forward pass time and the time required for data preprocessing. This table specifically presents the forward pass time to allow for a more detailed analysis of model performance. *For PROTAC calculations, 40 conformations were used for each test sample.

Time (s)	w/o GPU	w/ GPU
PROTAC*	4.79	1.85
MG(D)	0.15	0.05

Responds to Reviewer #2

Q1: I co-reviewed this manuscript with one of the reviewers who provided the listed reports. This is part of the Nature Communications initiative to facilitate training in peer review and to provide appropriate recognition for Early Career Researchers who co-review manuscripts.

Response: Thank you for your efforts in co-reviewing our manuscript. We appreciate the feedback provided in the reports and will carefully address the concerns raised.

Responds to Reviewer #3

Q1: Xue et al. describe a novel ternary complex prediction deep learning method trained on thousands of non-degrader ternary structures in the PDB that predicts degrader ternary complex structures faster and more accurately than previous methods. The model uses SE(3)-equivariant encoder to distill information in feature and coordinate vectors. This information is decoded by a pocket points decoder to identify where the degrader binds. These points are then used to position the degrader and the second protein. While the study is very novel and highlights the potential for machine learning application in this field, a number of major points should be addressed or clarified before publication.

Response: Thank you for recognizing our work and providing valuable suggestions. We appreciate your positive feedback and will address each of your concerns in detail below.

Q2: The last section before the discussion describes the “Optimization of model designs”, where they refine the choice and hyperparameters of the final model. Surprisingly, it seems like, while they are refined using the training set, the final selection of these hyperparameters is based on evaluations conducted on the test set. If this is the case, this is a flaw: test set and validation set should not overlap. If this is not the case, it should be clarified. It would be best not to keep this critical information at the very end of the Results section.

Response: Thank you for raising this issue. Our initial goal was to demonstrate the effectiveness of the proposed decoder and perform ablation studies with different hyperparameters. However, your comment has made us realize that we should have used a validation set, rather than the test set, for these experiments. Following your suggestion, we have incorporated the previously discarded complexes as the validation set. Specifically, we introduced a new metric, the Validation Performance Score (VPS), to aid in selecting the best model. VPS is calculated by averaging the DockQ scores of the top-ranked prediction based on PAE and the best-performing prediction evaluated on the validation set:

$$VPS = (D_{top-1} + D_{best})/2$$

This metric allows us to jointly account for both the model's output confidence and its generalization ability on the validation set. The experimental results regarding different hyperparameters (hidden dimension, noise level, and seed number) were provided in Supplementary Tab. A.1. As is shown, the model's performance on the validation set aligns well with the test set, and the best model on the validation set also got the best performance on the test set. For the MG(D) task, we still used a single seed to balance between the performance and computation cost.

The above explanations and results on the validation set were added to the Supplementary Materials. According to your suggestion, we also moved this section “Optimization of model designs” forward, and modified Fig. 6 to Fig. 3 in the revised manuscript.

Revision:

A.1 Model selection on the validation set

For model selection during training, we evaluated model performance on a validation set comprising curated structures dissimilar to the training set but also not in the test set. After training, a single scalar Validation Performance Score (VPS) is calculated via an arithmetic mean of the DockQ scores for the top-ranked prediction based on PAE ($D_{\text{top-1}}$) and the best overall prediction (D_{best}):

$$\text{VPS} = (D_{\text{top-1}} + D_{\text{best}})/2 \quad (\text{A1})$$

where D represents the DockQ score between the predicted and ground-truth crystal structures. Despite its simple definition, VPS assesses both structure and confidence accuracy.

The model’s performance on the validation set is summarized in Supplementary Tab. A.1. Unlike the PROTAC, the performance on MG(D) plateaued with increasing seed numbers. To balance performance and computational cost, we used a single seed for MG(D) evaluation, resulting in $\text{VPS} = D_{\text{top-1}} = D_{\text{best}}$.

Supplementary Table A1 Results on the validation set for model selection. The highest VPS for each hyper-parameter is marked in bold. ‘=’ denotes $D_{\text{best}} = D_{\text{top-1}}$ for MG(D)s, because only one seed is used for evaluation.

	PROTAC			MG(D)			Mean VPS
	$D_{\text{top-1}}$	D_{best}	VPS	$D_{\text{top-1}}$	D_{best}	VPS	
dim64	29.99	68.17	49.08	10.97	=	10.97	30.03
dim128	31.49	66.60	49.05	10.14	=	10.14	29.59
dim256	39.97	68.40	54.19	22.56	=	22.56	38.37
noise1	25.32	70.19	47.76	22.01	=	22.01	34.88
noise2	39.97	68.40	54.19	22.56	=	22.56	38.37
noise3	41.43	67.46	54.45	19.72	=	19.72	37.08
seed1	33.93	33.93	33.93	22.56	=	22.56	28.25
seed5	29.09	45.58	37.34	22.71	23.3	23.30	30.32
seed10	37.02	56.86	46.94	22.83	23.63	23.63	35.29
seed20	39.42	61.12	50.27	22.89	23.82	23.82	37.05
seed30	38.93	68.19	53.56	22.86	23.89	23.89	38.73
seed40	39.97	68.40	54.19	22.73	23.94	23.94	39.06

Q3: The authors should explain why a pocket points decoder is needed to superimpose the PROTAC back onto the protein when the binding pose of the unlinked ligands are already available. In those cases, the pocket points are known, and the model only adjusts the conformation of the PROTAC in the encoder to fit into the complex. The decoder might be more relevant for molecular glues complexes if the binding site on the target protein is not known. However, the performance for this is poor.

Response: Thank you for the valuable suggestion. For the PROTAC task, the decoder does not need to predict the pocket points, as these can be directly obtained from the unbound structures (described at the end of Methods.Decoder). However, the decoder is still required for generating PAEs, which are crucial for model evaluation. Therefore, for PROTACs, only PAE queries are input to the decoder, while for MG(D)s, both pocket point and PAE queries are used. It is worth noting that, due to the Transformer architecture’s inherent ability to handle variable numbers of input queries without requiring architectural modifications, we can use the same decoder for both PROTAC and MG(D), simplifying the model design and implementation. We have updated the description of model architectures, figure captions to make it clearer in the manuscript.

Revision:

313 and p2 are weight-shared. Following the encoding stage, the conformation of the small molecule
314 is utilized as the final conformation. The final ternary structure is generated based on this
315 predicted conformation and pocket points. For PROTACs, the pocket points are derived from
316 unbound structures, while for MG(D)s, these points are predicted by the proposed query-based
317 pocket points decoder (PPPD). With these information, we can rigid align the ligand and p2
318 back to p1 for form the final structure. The PPPD will also predict an alignment error for this
319 predicted structure. Notably, benefiting from the Transformer architecture's inherent ability
320 to handle variable numbers of input queries without architectural modifications, the proposed
321 PPPD architecture is unified for both PROTAC and MG(D). This simplifies the model design
322 and implementation. For PROTACs, only PAE queries are input to the decoder, while for
323 MG(D)s, both pocket point and PAE queries are used.
324

Q4: Many (most?) of the compounds in the training sets are probably molecules from the crystallization buffer and are not functionally relevant. This point should be evaluated and discussed.

Response: Thank you for highlighting this important point. It is true that many ligands in the PDB are crystallization buffers, which often appear across numerous PDB entries. Indeed, we did observe this phenomenon and thus manually excluded widely occurring ligands, including ACT, GOL, PEG, SO4, TRS, XYP, BME, EDO, PG4, and PG5.

Additionally, we notice that AlphaFold3 provides a more extensive list of crystallization aids. To assess the impact of a stricter exclusion criterion, we retrained our model by removing agents included in the AlphaFold3 list. The figure below demonstrates that the new results are comparable to those obtained from our original dataset. Based on this observation, we have decided to retain the original dataset while ensuring that the exclusion process is clearly outlined in the manuscript.

Revision:

of 0.26. Despite this, these structures were manually included in our databases due to their
significance. Additionally, many ligands in the PDB are crystallization buffers that frequently
appear across numerous PDB entries and are not functionally relevant. To address this, we
manually exclude commonly occurring ligands such as ACT, GOL, PEG, SO4, TRS, XYP,
BME, EDO, PG4, and PG5 from the dataset. The culmination of our efforts resulted in a
comprehensive structure collection dataset comprising 22,303 complexes.

733
734
735
736
737
738
739

Q5: Some of the data are not properly presented in figures. Some data seem to be missing in Figure 3A,B,C where the bottom plots only show 13 data points or less for 22 examples and it is unclear whether data from old benchmark or new benchmark are used (22 vs 43 complexes). Also, in the supplementary data, there a big difference between the first and second benchmark set where 6BOY initially gets a score of 0.7 and then 0.29: these differences are not

addressed in the text and it is unclear which data are plotted. This questions the reproducibility of the model and whether the author initially trained the model with the complexes that were later added to the second benchmark dataset. I would recommend the authors to run a reproducibility study to evaluate what the model's performance is over n of 3 or 5. Furthermore, 182 complexes are filtered out based on similarity to MG(D) complexes but are not added to the benchmarking dataset similar to the PROTAC dataset. This would increase the test set and would have allowed for a validation set. Why was this not added?

Response: Thanks for your questions and suggestions. We addressed each of the points below:

- 1) Different number of points in Figure 3 A, B, C.** This is because the authors of the compared methods such as Method 4B and the RosettaDock-based method only conducted evaluation on the subsets of the 22-example benchmark. For example, Method 4B was only tested on 6 PROTAC-induced complexes, so there are just 6 points in the corresponding plot. Please note that all data are from the 22-complex benchmark, which is the most comprehensive benchmark we can find from existing literature. We have now taken your suggestion into consideration and further clarified the figure by showing the sample points in the box plots in the revised manuscript.
- 2) Inconsistent results in supplementary tables.** We apologize for the discrepancy in the supplementary tables. This was an oversight; the supplementary data inadvertently contained outdated results from an earlier stage of our work. The correct, updated results for the 43-complex benchmark were inadvertently omitted. As these complexes now constitute our validation set, their performance metrics have been moved to a validation performance table in the Supplementary Materials, as described in our reply to Q2.
- 3) Reproducibility study.** We appreciate the reviewer's recommendation for doing a reproducibility study. Following this suggestion, we have retrained our model over five times with different random seeds (0, 1, 2, 3, and 4) for both model initialization and data sampling. The results, including mean, standard deviation, and min/max, are now provided in the Supplementary Materials. The standard deviations are 3.50 for PROTAC top-1 DockQ, 1.36 for PROTAC best DockQ, and 1.47 for MG(D) top1 DockQ. The largest variation was observed for PROTAC top-1 DockQ, ranging from 28.02 to 38.05. The performance metrics reported in the main text and figures now reflect the mean values from these five runs. Specifically, the reported best DockQ values are now 65.45 for PROTAC and 21.11 for MG(D). We hope this addresses the reproducibility concern and provides a more robust evaluation of our model's performance.
- 4) 182 complexes in MG(D).** We have taken your suggestion into account and use these complexes as a validation set. The performance on these complexes were reported in the validation performance table in the Supplementary Materials, as described to Q2.

Revision:

Supplementary Table A2 Reproducibility study of model performance across five training runs with different random seeds. Reported are the mean, standard deviation, minimum, and maximum values for key performance metrics.

	PROTAC		MGD
	Top1 DockQ	Best DockQ	Top1 DockQ
Seed0	29.54	67.21	22.34
Seed1	28.02	65.73	18.89
Seed2	38.05	64.74	22.61
Seed3	31.22	66.23	21.85
Seed4	33.5	63.22	19.85
Mean	32.07	65.43	21.11
Std	3.50	1.36	1.47
Min	28.02	63.22	18.89
Max	38.05	67.21	22.61

Q6: There is a focus on the protein-protein complex performance scores, but not on the PROTAC or MG(D). When looking at the figures for the predicted complexes, the PROTACs and MG(D) are predicted far from the pose in the crystal structure. The authors should provide data on the accuracy of the predicted PROTAC and MG(D) binding poses. This is only a suggestion, but for PROTACs, the authors may consider keeping the bound ligand(s) rigid during conformational searches and only keep the linker atoms flexible. This would improve the superimposition of the PROTAC back into the proteins. Currently based on the accuracy of the prediction of the compounds DeepTernary seems more a binary complex prediction than a ternary complex prediction.

Response: Thanks for your comments. Although the DockQ score only measures the protein-protein complex docking performance, it relies on the awareness of the ligand. As we illustrated in Figure 3f, PDB ID 6W7O and 6W8I have the same protein pairs, but the corresponding structures are kind of different due to the difference of PROTAC molecules. The examples BRD4-VHL and CRBN-BTK in the BSA analyses also support this observation.

Per your suggestion, we have now quantified the accuracy of our ligand pose predictions using RMSD relative to the crystal structures. We calculated both the RMSD of the top-ranked prediction (based on PAE) and the best RMSD achieved across all generated conformations with different random seeds. Our results in the following figure show that the mean top-1 RMSD for PROTACs is 3.43 Å, with 43% of predictions achieving an RMSD < 2 Å. In contrast, MG(D) exhibit a significantly higher mean top-1 RMSD of 13.14 Å, with only 1.9% of predictions below 2 Å. These results underscore the substantial challenge in accurately predicting MG(D) binding poses and highlight the need for further methodological development in this area.

We appreciate the suggestion to constrain the bound ligands during PROTAC conformation searches. We have explored this approach using `rdkit.AllChem.ConstrainedEmbed` function, as detailed in the `predict_btk.py` file in our GitHub repository. We found that it can only constrain one end and will fail on some complexes. We will be actively investigating alternative strategies to improve the accuracy of PROTAC and MG(D) pose prediction, and any further suggestions are welcomed.

Q7: The authors should comment on why PROTACs perform better than MG(D) (DockQ score average of 0.70 and 0.10 respectively) while the training set is skewed towards small molecule complexes.

Response: Thank you for the suggestion. Actually, the average DockQ for the MG(D) benchmark is 0.2, with 23% of the complexes predicted to have acceptable results (DockQ > 0.23). Still, it is far away from the performance of PROTAC. We think this is mainly due to the different docking settings between PROTAC and MG(D)s. For

PROTACs, both the pockets of the proteins and the two ends of PROTACs are known, making the searching space smaller. In contrast, MG(D)s present a much more complex challenge: both the small molecular and protein 2 can potentially bind to a wide range of locations around protein 1, and there is not any prior knowledge of the pocket. This big search space makes the model struggle to accurately predict the final ternary structure. Moreover, unlike PROTACs, MG(D) ternary complexes exhibit more complex interactions. In PROTACs, a small molecule acts as a linker, bridging two proteins, each of which can independently bind to it. MG(D)s, however, can involve simultaneous interactions between a protein, a small molecule, and another protein. This added complexity makes understanding MG(D) interactions more challenging.

Q8: I would recommend the authors to look at other small molecule complex prediction software such as Chai-2 and AF-3 to benchmark their model for MG(D) prediction.

Response: Thanks for the suggestion. We further tested AF-3 and Chai-1 in the revision. Please note that the training data for AF-3 and Chai-1 differ from ours—AF-3 uses PDBs released before September 30, 2021, while Chai-1 applies a cutoff date of January 12, 2021. Both training sets contain our test samples. To ensure a fair comparison, we first filtered out PROTAC and MG(D) test clusters that were included in AF-3 and Chai-1’s training data. For PROTACs, three test complexes were not part of AF-3’s training set: **7JTO_L_B_VKA**, **7JTP_L_A_X6M**, and **7Q2J_C_D_8KH**. We then performed MSA and template searches (to align with our unbound prediction setting as much as possible) and generated 40 predictions for each complex using different random seeds. The best DockQ scores for these three complexes were **0.12**, **0.25**, and **0.48** with AF-3, compared to **0.56**, **0.67**, and **0.53** with our model, indicating a clear decrease in performance of AF-3. It is also worth noting that the AF-3 trained on other 19 test complexes while our model never seen PROTAC-like structures during training. There is another complex, **7KHH_C_D_WEP**, that is not trained with Chai-1. Although Chai-1 could predict this complex in a high accuracy, up to 0.81 DockQ, it performs worse on other PROTAC complexes.

For MG(D)s, unfortunately, all test set clusters were included in AF-3’s training data. This means that either the test set was directly part of the training set or at least one sample in the training data was highly similar to a test complex. Consequently, AF-3 achieves a high mean DockQ on the MG(D) test set, 0.52 with one seed number. There is only one MG(D) complex **7LRD_B_A_X5M** that is not trained by Chai-1. It yields a DockQ of only 0.009, while our method achieves 0.014, on this sample.

The results here indicate that despite the greater resources used to train Chai-1 and AF-3, their poor performance on unseen PROTACs and MG(D)s highlights the generalization potential of our model.

Supplementary Table A3 Comparison with Chai-1 and AlphaFold3 (AF-3). Our DeepTernary illustrate great generalization to unseen PROTAC and MG(D). – indicates that AF-3 is trained on this complex cluster.

Type	Complex ID	Chai-1	AF-3	DeepTernary
PROTAC	7JTP_L_A_X6M	0.40	0.25	0.67
	7JTO_L_B_VKA	0.04	0.12	0.56
	7Q2J_C_D_8KH	0.06	0.48	0.53
	7KHH_C_D_WEP	0.81	-	0.38
MG(D)	7LRD_B_A_X5	0.009	-	0.014

Q9: It is said that the training set is made available, but it is hard to find on the github page. I would highly recommend for the authors to provide at least a list of PDB codes + chains in the supplementary data as csv file for the training set, so other researchers can use the training set for model development. Since, in the second benchmark, several ternary complexes are missing (8BB4, 8BEB, 8DSO, 8FY0, 8FY1, 8FY2, 8PC2, 8QJR, 8QVU, 8QW6, 8QW7, 8R5H) this questions whether these might have ended up in the training dataset causing data leakage.

Response: Sorry for the missing description of the training list. We have uploaded the dataset list with PDB ID, chains, and ligand CCD names and the cluster results under the `data` folder of the codebase (data/PROTAC/train_clusters.json). We have also updated the README file to include the description. As for the PROTAC PDB IDs you listed, none of them were in our collected dataset, primarily due to their high R-value free. For example, both 8BB4 and 8BEB have an R-value free of 0.27, exceeding our 0.26 cutoff. 8R5H, was excluded because its structure was determined using electron microscopy, whereas we restrict our dataset to X-ray diffraction structures.

Q10: Since BSA is highlighted late in the paper, it would be nice to add analysis of BSA of the training set.

Response: Thanks. As suggested, we have calculated BSA for all complexes in our training set. The results are displayed in the figure below, where the BSA of the test set, as well as the analyses of the BRD4-VHL and CRBN-BTK pairs, are also added for comparison. Interestingly, we found that the BSA distribution in the training set is not identical to that of the test set. The training set exhibits a slightly lower average BSA, with a peak around 1000 Å². This finding suggests that our model is not simply overfitting to the training set. Instead, it appears to be learning more generalizable principles of ternary complex formation. This supports our hypothesis that the model is capturing underlying interaction principles and has the potential to predict degradation potential based on the BSA calculated from the predicted structure. We have added these results and the above discussion to the supplementary information.

Q11: Line 370: The text mentions only 22 complexes for the test set but then in methodology this is increased to 43 complexes, I think the text here should reflect that.

Response: Thank you. We provide more explanations here. The 22 complexes were reported in a previous study (i.e. the BOTCP method), which is the most comprehensive benchmark available to date. We treated this as our test set. During the similarity clustering process of our collected dataset, we found that there are some complexes not included in this test set. We manually checked these new complexes and found all of them are PROTAC-induced structures. Combined with these newly retrieved complexes, we finally landed on a 43-complex benchmark. However, as you suggested in Q2, we have adopted these new complexes as a validation set. So, there will be only the 22-complex test set in the manuscript. We have updated the manuscript to make this point clearer.

Q12: Line 406: “The acceptable prediction appeared within the top-ranked results” what is considered the top-ranked? Top 10, top 5, top 3? Generally, the sentence “Importantly, the majority of acceptable predictions appeared within the top-ranked results, with approximately half deemed reasonable” is vague, what is considered acceptable and reasonable?

Response: Thank you for pointing out the ambiguous expression. To clarify, we generated 40 predictions for each test PROTAC using different random initial conformations. These predictions were then ranked by the Predicted Alignment Error (PAE). We defined the metric “acceptable rank” as the order of the first acceptable prediction (DockQ > 0.23). As shown in the middle panel of Figure 3a, our model typically ranks an acceptable complex at an average position of 4.06, indicating that there is generally at least one acceptable prediction within the top four results. Besides, Figures 3b and 3c display the success rates under multiple quality criteria. We have now added explicit definitions of “CAPRI criterion” and “RMSD < 10 Å” in the Metrics section and inserted pointers in the Results section. As the success rate increases to 50%, roughly half of the predictions meet the “acceptable” quality threshold based on the selected criteria. We have updated this Results section as well as the Metrics section in the manuscript to make it clearer.

Revise:

using different random seeds, each of which was input into our model. To estimate the prediction quality, we introduced a predicted aligned error (PAE), allowing us to rank the predicted results and select the most confident output. With an average rank of 4.06 under 40 seeds (Fig. 4a, middle panel), DeepTernary reliably generated acceptable predictions (DockQ > 0.23). In another words, the majority of acceptable predictions appeared within the top-4 results. To compare with existing methods, we also calculate the prediction success rate for each complex based on another two criteria: CAPRI criteria and RMSD < 10 Å (Fig. 4b and c). As we can see, DeepTernary significantly improve the success rate to around 50%, which means for most of the test complexes, more than half of the predictions are above acceptable quality. Since the ground-truth structure is typically unavailable in practice, distinguishing between better or worse output structures can be challenging in the absence of a reliable scoring or ranking

Q13: How different are the PAE scores between the most accurate prediction and other predictions? Are these small difference or big difference? It would be nice to see that PAE score of the most accurate prediction can indeed be used as a confidence score and a filter. I would recommend including a plot to show the correlation between the DockQ scores and the PAE scores to validate that PAE can be used as a confidence score for the predicted complex.

Response: Thank you for the question and the suggestion. We have now added a supplementary to plot the correlation between the DockQ scores and the PAE scores for PROTACs and MG(D)s. We displayed the 40 predictions for each complex, as well as the top-1 and best predictions for all complexes. In general, our results reveal a strong negative correlation—complexes with lower PAE scores tend to have higher DockQ scores, indicating more accurate predictions. Although a few cases (e.g., 6BN7_B_C_RN3, 6BOY_B_C_RN6, 6HAX_B_A_FWZ, 6HAY_F_E_FX8, 6W7O_C_A_TL7, and 6ZHC_A_D_QL8) do not show a clear correlation, the overall trend supports the idea that the PAE score can serve as a reliable confidence measure and effective filter for selecting high-quality predictions in drug discovery applications.

Supplementary Figure A3 Correlation of PAE and DockQ score for predicted ternary complexes.

a, Intra-complex correlation for PROTACs. Scatter plots showing the relationship between PAE and DockQ scores for 40 initial conformations of each PROTAC test complex. Each point represents a single conformation. The majority of complexes exhibit a negative correlation, indicating that lower PAE values tend to correspond to higher DockQ scores. This suggests that PAE can serve as a useful indicator of prediction accuracy within a given complex. **b**, Scatter plot showing the relationship between PAE and DockQ scores for the best-predicted conformation (highest DockQ) of each PROTAC test complex. The plot demonstrates a clear trend: complexes with lower PAE scores (< 4) tend to have higher DockQ scores (> 0.5), further supporting the use of PAE as a confidence metric. **c**, Across-Complex Correlation for Top-1 (lowest PAE) PROTAC Predictions. While some false positives are present, the overall trend remains a negative correlation. **d**, Correlation for MG(D)s. The plot demonstrates a clear negative correlation, indicating that lower PAE values are associated with higher DockQ scores, similar to the trend observed for PROTACs. This suggests that PAE can also be used as a confidence metric for MGD predictions.

Q14: Line 403: Rank of what? The closest prediction to the crystal structure? How far off was the number one prediction to the crystal structure? How different were the predictions among each other for one complex, was there convergence to one ternary complex conformation if sampled enough PROTAC conformations?

Response: Thank you for your questions. In our study, we obtained predictions using different initial PROTAC conformations. The “rank” here refers to the ordering of these predictions based on their PAE scores, where a lower PAE indicates a prediction that is more similar to the crystal structure. In other words, the top-ranked (number one) prediction is the one with the lowest PAE and is assumed to be the closest to the crystal structure.

In addition, to address your specific queries, we provided more clarifications below:

1) Difference to the Crystal Structure: We compared the top-1 prediction (i.e., the one with the lowest PAE) to the crystal structure using the DockQ score. We provide a table in the manuscript that tracks how this top-1 DockQ score changes as we increase the number of sampled conformations. Initially, as more conformations are sampled, the DockQ score improves, indicating a closer match to the crystal structure.

2) Convergence Among Predictions: The data shows that the improvement in the DockQ score plateaus after about 20 conformations. This plateau suggests that the predictions converge to a stable ternary complex conformation, and additional sampling beyond this point does not further improve the quality of the top-1 prediction.

Overall, our method converges to high-quality predictions far more efficiently than existing approaches, which typically require sampling thousands of conformations.

Supplementary Table A5 Top-1 DockQ scores with different numbers of sampled random conformations on the PROTAC test set. The results are tested on the same checkpoint.

Sample Number	Top-1 DockQ
1	0.33
10	0.37
20	0.40
40	0.40

Q15: Line 404: Ligands are conformationally sampled, was this done for the training step as well?

Response: Yes, we applied conformational sampling to the ligands during the training phase as well. For each training ligand, we generated and stored 50 random conformations. During training, one of these 50 conformations was randomly selected for each instance. We have now included this detail in the manuscript.

Revised:

Training and inference

During the training process, protein structures were derived from bound structures and ligand conformations generated by the RDKit toolkit [42]. For each training ligand, we pre-generated a pool of 50 random conformations. In each training iteration, protein1 or protein2 was randomly fixed, while the other protein and a randomly selected conformation from the ligand’s 50-conformation pool were subjected to random rotations and translations from their original positions. Coordinates were normalized before being input into the model to stabilize

Q16: Line 483: “The PAE matrix in the top right corner further highlights AF3’s lower confidence”: it would be nice to define somewhere in the text what is considered a confident PAE score.

Response: Thank you for the suggestion. The PAE in AF3, measured in Ångströms (Å), represents the expected positional error between two residues in the predicted structure. Typically, an AF3 PAE value exceeding 15 Å is considered indicative of a less confident prediction. In the specific case of the interaction between p1 and p2, the PAE values are approximately 20 Å, which highlights the lower confidence in this predicted interaction. We have incorporated this explanation into the manuscript to ensure clarity for the reader.

Revised:

tions, as we did with EquiDock. Fig. 5b illustrates a prediction for PDB ID 7BQU, where AF3, thought better than EquiDock, performs significantly worse than DeepTernary, which predicts a structure closest to the co-crystallized ground truth (green vs. gray). We show the PAE matrix in the top right corner. The PAE in AlphaFold, measured in Ångströms (Å), represents the expected positional error between two residues in the predicted structure. Typically, an AF PAE value exceeding 10 Å is considered indicative of a less confident prediction. In this prediction, the PAE values are approximately 20 Å, highlights AF3's lower confidence in this predicted interaction.

Q17: Line 527: “This suggests that the large, well-folded domains in Group 1 complexes involve more complex binding rules, which may not be adequately covered by the training set”: it would be nice if this analysis was performed on the training set to further support this conclusion. It may well be the other way around, where the PDB has more ternary complexes with folded regions and molecules than small recognition motifs.

Response: Thanks again. We have now included an analysis of the protein length in the training set for the MG(D) model. Specifically, we have added a figure to the supplemental materials, which presents: (a) the joint distribution of sequence lengths for the pairs of interacting chains, and (b) the overall distribution of sequence lengths across all chains in the training set. This reveals that the training set contains a substantial number of proteins with folded regions (sequence lengths ranging from 200 to 600 residues), as well as a significant number of shorter proteins (20 to 40 residues). Furthermore, the joint distribution shows that the most frequent type of complexes in the training set involves a longer protein interacting with a shorter protein, which is indeed akin to the Group 2 complexes in our test set. This analysis supports our conclusion that the more complex binding rules observed in Group 1 complexes, which often involve interactions between larger, well-folded domains, may not be as well-represented in the training data. We have incorporated this analysis and discussion into the manuscript and the supplementary.

Q18: Line 538: “They also showed that lower K_{LPT} corresponds to higher ternary complex binding affinities and, consequently, higher degradation potency”. Do the authors mean “lower BSA” instead?

Response: Thank you for the question. We have modified this sentence to avoid confusion. The K_{LPT} represents *equilibrium dissociation constant* by Wurz et al.’s work. The lower K_{LPT} corresponds to higher BSA and higher degradation potency. We have now revised the manuscript to use this term instead of binding affinity.

Figure 7d in Wurz et al.'s work demonstrates that lower K_{LPT} values correspond to higher BSA.

Revised:

Experimental work by Wurz *et al.* [19] has demonstrated a strong correlation between the total buried surface area (BSA) of PROTAC-mediated ternary complexes and the *equilibrium dissociation constant* (K_{LPT}) for VHL-PROTAC-SMARCA2. Their findings revealed that BSA has a negative correlation with $\ln(K_{LPT})$, while a lower K_{LPT} corresponds to higher degradation potency. In other words, a higher BSA corresponds to higher degradation potency. To test whether our predicted ternary structures could reflect this relationship, we calculated the

Q19: Line 540: How many PROTACs were in this dataset? How strong is the correlation? What is the R^2 ?

Response: In the work by Wurz et al., they include 11 PROTACs for SMARCA2. However, for the correlation analysis with BSAs, they utilized 9 compounds, excluding compounds 8 and 10. While the authors noted that no initial model for compound 8 passed their selection criteria, the reason for excluding compound 10 was not explicitly stated. Following their results, we used these compounds in our correlation analysis (resulting in 9 points in Fig. 5b). The R^2 value is 0.2964, not very strongly correlated. However, it is important to note that our analysis does not aim to predict BSA based on $\ln(K_{LPT})$. As we discussed in the manuscript, the relationship between BSA and degradation potency is not linear or even not monotone.

Q20: Line 570: The authors conclude that the differing trends likely stem from intrinsic differences between E3, but as reported early, the short linker caused clashes between the proteins. What is the conclusion if you only look at predictions where no clashes are found? If you zoom in on PROTAC 5-11 and plot the correlation between BSA and $\ln(\text{DC}_{50})$ would you see a similar correlation/trend to the Wurz et al paper? It would be good to specify whether the authors recommend for future groups to use the BSA metric as a virtual screening tool and explain why or why not.

Response: Thank you for the insightful question. When we restrict our analysis to predictions without clashes (PROTAC 5-11), our findings align with the original paper (Zorba et al.). That is, lower BSA has lower DC50/high degradation potential. However, this observation conflicts with Wurz et al.'s paper (please check our response to Q18). By unifying the different conclusions from Zorba et al. and Wurz et al., we derived that instead of a simple linear correlation, there exists an optimal BSA range—approximately 1175 to 1422 Å²—that is associated with enhanced degradation potency. Based on this observation, we suggest that BSA can indeed serve as a useful metric in virtual screening and recommend future studies to target the optimal BSA range rather than assuming a monotonic relationship. This approach could help guide the design and selection of PROTACs with a higher likelihood of achieving productive degradation.

Q21: Does the model take into consideration induced fitting of the complexes as observed in PDB: 6HAX.

Response: Thank you for the question. Our current model treats proteins as rigid bodies, using the conformations provided by existing PDB structures. Therefore, it does not account for induced fit effects. We are interested in this scenario and will investigate it more in the future.

Q22: I would recommend using swarm plots for all boxplots to show how many test cases there are.

Response: Great suggestion! We have updated the manuscript to incorporate swarm plots (i.e., displaying individual data points) on all boxplots. This enhancement clearly indicates the number of test cases in each plot and improves overall data transparency.

Q23: Figure 1:

- B) Disassemble implies that the parts from ternary complex were used for training rather than matched to the monomers of the respective proteins which has been done for the PROTAC complexes. So, would recommend including this in the figure to avoid confusion.
- C) Is the protein missing for the purple stars?
- C) The step where conformations are generated for the PROTAC/MG(D) is missing in the figure

Response: Nice suggestion, again. We have updated Figure 1b to add a note about disassemble parts, updated 1c to add the missing blue protein. We added a note to 1c to indicate that the conformation of PROTAC/MG(D) is initialized by RDKit and gradually updated in the SE(3)-Equivariant Encoder, as reflected by the slight change in ligand conformation.

Revised:

Q24: Figure 2:

- E) the text in the figure legend does not match the figure, no colouring for clusters are visible.

o “e, Proteome-wide view of the collected dataset. ESM-1b [38] sequence embeddings for the two proteins in each complex are calculated and concatenated. This is followed by two-dimensional (2D) Uniform Manifold Approximation and Projection (UMAP) and Density-Based Spatial Clustering of Applications with Noise (DBSCAN), with clusters distinctly marked using a qualitative color scale. Similar complexes to PROTACs- and MG(D)s-involved ternary structures are denoted as red and green square points, respectively. “

Response: Thank you for catching this error. The panel E indeed doesn't have the cluster colors. Our initial version of the plot did apply different colors to the clusters (as shown below). However, we found that the cluster information has already effectively reflected in the spatial positions, so we have removed the cluster colors to avoid duplication. We have deleted this description from the caption.

Q25: Figure 3:

- Are these boxplots scores for the best predicted complexes for each of the complexes or average for each test case?
- F) In the crystal structure PDB: 6W8I, there are three different poses for the complex, which one was used for comparison?
- F) the last panel shows that the PROTAC does not overlap with the cIAP ligand.
- D) what causes the big difference in DockQ score between BIRC2 and CRBN?
- The scores in the supplementary for the CRBN complexes are 0.70 and 0.71 DockQ, but the average here seems to be 0.50 which does not reflect the data. In supplementary table 2 the results are 0.62 and 0.29 which would make sense to result in 0.50. What caused the difference in result in supplementary table 1 vs supplementary table 2 for the same complex, for example 6BOY?

Response: We addressed each of your question below:

- Are these boxplots scores for the best predicted complexes for each of the complexes or average for each test case?
In the revision, the boxplots now show the mean performance across five independent runs (using different random seeds). For each complex, we first selected the best-performing prediction (e.g., the highest DockQ score from 40 predictions). We then calculated the mean of these best scores across all test complexes and finally averaged this mean across the five runs.
- F) In the crystal structure PDB: 6W8I, there are three different poses for the complex, which one was used for comparison?
The 6W8I is 6W8I_D_A_TKY, and we have updated the annotation to include the chain ids in the figure.
- F) the last panel shows that the PROTAC does not overlap with the cIAP ligand.
Yes, you are correct. As demonstrated in our response to Q6, our model exhibits limitations in accurately predicting precise ligand positions. We are exploring methods to improve the accuracy of ligand pose prediction in future work.
- D) what causes the big difference in DockQ score between BIRC2 and CRBN?
By adding the swarm plots to the boxplot, we can find that the observed difference in DockQ scores between BIRC2 and CRBN may be attributed, at least in part, to the limited number of CRBN complexes in the test set (only two). This small sample size can lead to higher variance and potentially less robust performance. It is possible that with a larger dataset of CRBN complexes, the performance of our model might be more consistent and comparable to its performance on BIRC2.
- The scores in the supplementary for the CRBN complexes are 0.70 and 0.71 DockQ, but the average here seems to be 0.50 which does not reflect the data. In supplementary table 2 the results are 0.62 and 0.29 which would make

sense to result in 0.50. What caused the difference in result in supplementary table 1 vs supplementary table 2 for the same complex, for example 6BOY?

We sincerely apologize for the discrepancy in the DockQ scores between Supplementary Table 1 and 2. During writing the manuscript, we got a better checkpoint but forgot to update all the tables. In the revised manuscript, unless explicitly stated, all reported results, including those in the supplementary tables, represent the mean values from five independent runs to ensure reproducibility. Thank you for bringing this to our attention.

Q26: Figure 4:

- I would recommend also using the high/medium/acceptable, RMSD and rank metrics for the molecular glue dataset. Along with MG(D) RMSD. The ternary complexes show quite a difference in RMSD in the figure specifically the MG(D) RMSD which would not be suitable for structure activity optimization if $>1.5\text{\AA}$
- B) include color label for crystal structure

Response: Thank you for the suggestions. The percentage of high/medium/acceptable and $\text{RMSD} < 10\text{\AA}$ are measured within the same complex with different predictions (from different input conformations). However, for MG(D), we only adopt one random conformation as input, so we can only calculate the overall success rate for the test set. Among the 94 test complexes, there are 37% satisfying the CAPRI criteria (high/medium/acceptable), 51% are predicted with $\text{RMSD} < 10\text{\AA}$. The MG(D) RMSD is 13.14\AA , with only 1.9% of predictions below 2\AA , indicating the very difficulty of this task. We have included these results in the revised manuscript and included color label for crystal structure in panel B.

Q27: Figure 5:

- B) What is the correlation R^2 ?
- C) How well does BSA reflect the DC50 values?

Response: Thank you for the questions. As mentioned in our reply to Q19, the R^2 value between BSA and $\ln(K_{LPT})$ is 0.2964, which indicates a modest association. It is important to note that our analysis was not designed to predict BSA based solely on $\ln(K_{LPT})$ values; rather, we aimed to explore the complex relationship between BSA and degradation potency. In our study, we combined estimated $\ln(\text{DC50})$ values from the work (Zorba et al.) which assigns $\ln(\text{DC50}) = 10$ for cases with no degradation, with data from the method (Wurz et al.). The resulting plot (shown below) demonstrates that while there is some correlation, the relationship is not linear or monotonic. Instead, the trend suggests that when BSA deviates from an optimal interval, $\ln(\text{DC50})$ increases—indicating reduced degradation potency. This analysis implies that an optimal range of BSA is more predictive for the degradation potency of PROTACs.

Responds to Reviewer #4

Q1: In this work, the authors propose a deep learning model that leverages SE(3)-equivariant graph neural networks (GNN) to predict the ternary protein complexes that are involved in target protein degradation. The proposed model is evaluated on the MG(D) benchmark dataset, which is a dataset of 22 known PROTAC-induced ternary structures proposed by Rao et al. in "Bayesian optimization for ternary complex prediction (BOTCP)". The authors compared their method to existing methods, such as FRODock, RosettaDock, and BOTCP. The results show that the SE(3)-equivariant model outperforms the baseline methods in predicting ternary complexes, concluding that SE(3)-equivariant GNNs can be a superior and fast alternative to docking methods.

The authors take great care in avoiding any data leakage between the training and test data. They also provide a detailed description of the model architecture and the loss functions used for training. Nevertheless, the authors could improve the paper by refactoring the order of the sections and the figures reporting their results, as well as adding documentation to the provided code. It currently appears that it would be very hard to reproduce the results of the work, so adding detailed guides on how to re-run the code and reproduce the results would be beneficial.

Response: Thank you for the constructive suggestions. We have reorganized the order of the sections and figures to make the paper easier to follow, added more details about the figure annotations, refactored the public code base to introduce details about the dataset and code structure. We hope these efforts could be helpful for future protein degradation research.

Q2: As mentioned above, despite the authors providing a code repository (which is great), there is no clear guide on how to obtain the training data nor on how to actually train the model, should one want to reproduce the results. Also, it is very hard to navigate the directory tree of the project, as it lacks valuable READMEs on how to navigate the codebase.

Response: Thank you for the feedback. We have addressed these concerns by improving the organization and documentation of our code repository:

1) Training data accessibility: The training data, including PDB IDs, chain IDs, and ligand CCD names, have been uploaded to the ``data`` folder of the codebase. Additionally, we have provided a zip package for convenient downloading.

2) Training instructions: Our training codes are built upon two open-source frameworks, MMEngine and MMPretraining, for streamlining the training process. All training configurations are consolidated into a single config file (`deepternary/configs/protac.py`). To initiate training, users can simply follow the command provided in the README: ``bash tools/dist_train.sh projects/equibind/configs/protac.py 2``.

3) Codebase navigation: We have refactored the directory structure to improve clarity. All codes to run our method are now organized under the `./deepternary` folder, with config files, model definition, and training process. We hope these updates can make it significantly easier to navigate the codebase and reproduce the training procedure.

Revised:

TernaryDB

The collected TernaryDB dataset can be downloaded here.

The PROTAC, the `val.csv` and `test.csv` have three columns, they represent complex_id, unbound structure for p1 and p2, respectively.

The similarity clustered results for the training examples are also included in the corresponding folder named `train_clusters.json`.

File structure:

```
TernaryDB
├── MGD
│   ├── test.txt
│   ├── train_clusters.json
│   ├── train.txt
│   └── val.txt
└── PROTAC
    ├── test.csv
    ├── train_clusters.json
    ├── train.txt
    └── val.csv
```

Prediction

To perform evaluation, follow these steps:

1. Download the pre-trained checkpoint and PROTAC unbound structures from this link.
2. Unzip the folder and move it to the root directory of this codebase.
3. Execute the following command to evaluate the model:

```
# download the pre-trained checkpoints and unbound structures
wget https://github.com/youqingxiaozhua/DeepTernary/releases/download/untagged-4b1648709c8f32cf6d26/output

# unzip data
unzip output.zip

# evaluate on the PROTAC test set
python predict_cpu.py output/DeepTernary/PROTAC
```

Training

For training with 2 GPUs, use the command below:

```
bash tools/dist_train.sh deepternary/configs/protac.py 2
```

Codes for model and dataset are located under `deepternary/models/`, their definition is in the config files under `deepternary/configs/`.

Q3: It is unclear what are the input and output data for the model. As far as I understand, the encoder part already expects the docked structure of the two proteins, and then outputs the refined 3D coordinates of the ternary complex. How are the pocket points defined and fed to the decoder part? Are the "pocket queries" suggested by the user? The authors briefly attempt to answer these questions in lines 358-361, but I suggest to provide a more detailed explanation and to refer to later sections of the paper for more details. This could also be better clarified in the figures.

Response: Thanks for this question and the suggestions. We apologize for the confusion and have revised the manuscript to improve clarity.

For the MG(D) task, our model takes as input: **1)** the 3D structures of the two proteins and **2)** the ligand's SMILES representation. The model outputs are: **1)** the ligand conformation and **2)** the predicted relative positions of protein 2 and the ligand with respect to protein 1. This input-output relationship is illustrated in the right panel of Figure 1b:

The "pocket points" facilitate the docking of protein 2 and the ligand from their initial random positions onto protein 1 (as shown in Figure 1c, right panel). These points correspond to the coordinates of the residues within the binding pocket. The "pocket queries" are the input queries of the Decoder, functioning within the attention mechanism. Importantly, these queries are randomly initialized without user input and are learnable during training, as detailed in the *Model Architecture - Decoder* part of the Methods section.

For PROTACs, since the binding sites at both ends of the ligand are already known, the model does not need to predict pocket points. Instead, these points are directly computed from the unbound protein structures. In this case, the decoder is used exclusively for PAE prediction, and the PAE query serves as the sole input to the Decoder.

We have revised the manuscript to include a more detailed explanation of these concepts and have added some pointers to the Methods section.

Q4: How are the "groups of pocket points" defined? I believe the authors give an explanation in the "Model architecture" section, but it is not clear to me. I suggest to provide a more detailed explanation (or pointers) already when mentioning Figure 1, i.e., in the "Introduction" section.

Response: Thank you for the suggestion. The pocket points correspond to the residue coordinates within the binding pocket. Since both the target and binder have their respective binding sites, each protein-ligand interface is associated with two groups of pocket points (two sets of coordinates). For ternary complexes, there are two binding pockets, leading to a total of four groups of pocket points. To enhance clarity, we have revised the manuscript to use the term "two pairs of pocket points" instead. Additionally, we have added a pointer to its definition in the *Model Architecture* section for better accessibility.

Revised:

Specifically, for MG(D)s, we defined **two pairs of pocket points**: ($\mathcal{P}_{\text{lig}}, \mathcal{P}_{\text{p1} \rightarrow \text{lig}}$) and ($\mathcal{P}_{\text{p2}}, \mathcal{P}_{\text{p1} \rightarrow \text{p2}}$). The first pair represents the pocket points between the ligand and protein1, where \mathcal{P}_{lig} denotes the ligand pocket bound to protein1, and $\mathcal{P}_{\text{p1} \rightarrow \text{lig}}$ denotes the protein1 pocket bound to the ligand. Similarly, the second pair represents the pocket points between protein2 and protein1, with \mathcal{P}_{p2} denoting the protein2 pocket bound to protein1, and $\mathcal{P}_{\text{p1} \rightarrow \text{p2}}$ denoting the protein1 pocket bound to protein2. Their corresponding queries are matrices $\mathcal{Q}_{\text{lig}}, \mathcal{Q}_{\text{p1} \rightarrow \text{lig}}, \mathcal{Q}_{\text{p2}}, \mathcal{Q}_{\text{p1} \rightarrow \text{p2}}$, each row of which denotes the query of each node. In addition,

Q5: The authors are careful to exclude any PROTAC and molecular glue molecules from the training data. This means that the model is trained on solely out-of-distribution data only. Because of that, how can the model generalize to unseen PROTACs and molecular glues? Since the model seems to be able to predict the ternary complex with high scores, how do the train and test data distribution look like? How similar are they? How different?

Response: Thank you for your question. Our core hypothesis is that the fundamental physical principles governing atomic interactions are universal, whether they involve protein-protein, protein-ligand, or more complex ternary

systems such as those formed by PROTACs and molecular glues. We believe that a deep learning model, if trained effectively, can learn these underlying principles and generalize to novel systems. This is analogous to how AlphaFold can predict the structure of de novo designed proteins, even those with very low sequence similarity to native proteins.

To test this hypothesis, as described in the Methods section, we employed a strict clustering strategy based on a 50% sequence identity threshold. This means that every complex in our training set has least one chain that shares less than 50% sequence identity with any chain in the test set. This creates a significant difference between the training and test data distributions.

Although our model's performance is promising, there is still room for improvement in the MG(D) task. For example, on the MG(D) test set, where the binding site is unknown, the mean DockQ score is just above 0.2—significantly lower than the 0.6 achieved on PROTACs, where pocket points can be derived from unbound structures.

We believe that our train-test split strategy is crucial for accurately assessing the true generalization ability of our model. This is especially important for real-world drug discovery applications, where predicting the behavior of novel molecules is essential. Our results thus serve as a valuable benchmark for the potential of deep learning methods in this domain.

Q6: Line 357 mentions that the model parameters for the two proteins are shared in order to capture the symmetry of the protein complex. How is this beneficial? In case of PROTACs, POIs and E3 ligases can be quite different, so there would not be much "symmetry information" for the model to leverage. On the contrary, I would expect the model to learn different parameters for the two proteins. Could you provide more details on this choice? I do acknowledge that the model might be faster because of this design choice, but I would still expect a more detailed motivation from the authors.

Response: Thanks for your question and we apologize for the confusion. The "symmetry" we refer to is not related to structure similarity between the two proteins (e.g., the E3 ligase and the POI in a PROTAC complex). Instead, it refers to the ternary complex itself. Specifically, given a ternary complex composed of protein1, a ligand, and protein 2 (p1-lig-p2), the predicted structure should be invariant to the order of the proteins. In other words, the predicted structure should be the same regardless of whether the input is (p1-lig-p2) or (p2-lig-p1).

Our motivation for sharing parameters between the two protein encoders is not to assume structural similarity between the proteins. Rather, it's to encourage the model to learn generalizable features from protein structures that are relevant to the overall ternary complex formation. We hypothesize that these shared features will benefit the downstream prediction of ligand conformation and pocket points, regardless of the specific identities of the two proteins.

On the other hand, as you pointed out in the comment, the proteins in our test set are diverse, making it impractical to assign one encoder specifically to E3 ligases and the other to POIs. Instead, our approach encourages the model to extract fundamental structural patterns that apply broadly across different proteins.

In fact, to further enforce this symmetry, we randomly swap the order of p1 and p2 during training, as described in *Training and inference* part of the Methods section. This ensures that the model learns to treat the two proteins in a symmetric manner. We have revised the manuscript to provide a more detailed explanation of our motivation and the specific training procedures.

Q7: The evaluation metric referred as "High/Medium/Acceptable" is not clearly defined. I suggest to provide a more detailed explanation of this metric. I would suggest to add references to the "Evaluation metrics" section and provide a more detailed explanation of the "High/Medium/Acceptable" metric there maybe.

Response: Thank you for the suggestion. We have addressed this in several ways. First, we have renamed the "High/Medium/Acceptable" metric to "CAPRI acceptable" to improve clarity. Second, we have added a clear pointer to the "Evaluation Metrics" section of the manuscript, where we provided a more detailed explanation of the CAPRI acceptable criteria. Finally, we have updated the figure legends to reflect the change to "CAPRI acceptable" and to ensure consistency throughout the manuscript. These changes should provide a more transparent and well-defined understanding of our evaluation process.

Revised:

To compare our methods with previously published approaches, we also calculated the fraction of acceptable predictions and compared them with other methods. It is worth noting that the criteria for an "acceptable" prediction vary across different studies. We categorize these criteria as follows:

DockQ > 0.23: This threshold indicates a quality prediction based on the DockQ scoring system.

CAPRI criteria: Derived from the Critical Assessment of Predicted Interaction (CAPRI) [57], predictions are classified into High, Medium, or Acceptable. This criterion has been employed to assess the quality of PROTAC-induced complex predictions, as used by Drummond *et al.* [29].

RMSD < 10 Å: This criterion involves calculating the C α RMSD and is commonly used as the upper limit for an "acceptable" pose in protein-protein docking contexts. It is straightforward and easy to compute.

For PROTACs, given the model's generation of multiple predictions from varying initial conformations, we employ the **Rank** metric, following existing methods. This metric is determined by sorting predictions based on their predicted alignment error (PAE) and identifying the rank of the first prediction achieving a DockQ score greater than 0.23.

By applying these metrics, we ensure a robust evaluation of our model's performance in predicting ternary complex structures.

Q8: The IEGMN architecture (which the authors cleverly customized for this work) is well detailed in the "Model architecture" section. However, it is already mentioned before in the "Optimization of model designs" section, without any reference nor pointer to later sections, which is a bit confusing. I suggest to re-order the sections as to first describe the model architecture and then discuss the optimization of the model design, and finally how it performs in terms of computation time.

Response: Thanks for your suggestion. We re-ordered the sections as you suggested and added more pointers in the manuscript to ensure a smooth reading flow and guide the reader to the relevant information.

Q9: Section "Data collection and filtering" under "Methods" is not detailed enough for reproducing and obtaining the training data from the raw PDB structures. Also, it doesn't add significant more details than what already explained in the "Results" section. I suggest to provide a more detailed explanation of the data collection process and eventually pointers to the code repository.

Response: Thanks for your suggestion. The data collection is completed by multiple steps. After reviewing our collection process, we supplemented the clash detection and filtering crystallization buffers to the section "Data collection and filtering" under "Methods". We also published the full data list in the code repository and followed your suggestion to add a link pointer to the repository in the manuscript.

Revised:

In our final step to validate meaningful protein-ligand interactions, we implemented a two-tiered filtering approach. First, we excluded complexes where the ligand established fewer than three contacts with the protein, defined as ligand atoms positioned within 4 Å of any protein atom. Second, we removed complexes exhibiting steric clashes, identified as any heavy atom pair (one from the ligand and one from the protein) separated by less than 2 Å. While this stringent criteria led to the exclusion of some well-characterized PROTACs and MG(D)s, such as PDB ID 6HAX [51] ($R_{\text{free}} = 0.268$), which marginally exceeded our 0.26 threshold, and PDB ID 6BN7 [52] (chain B ligand clash of 1.97 Å), these structures were manually curated and retained in our database due to their established significance. Additionally, many ligands in the PDB are crystallization buffers that frequently appear across numerous PDB entries and are not functionally relevant. To address this, we manually exclude commonly occurring ligands such as ACT, GOL, PEG, SO4, TRS, XYP, BME, EDO, PG4, and PG5 from the dataset. The culmination of our efforts resulted in a comprehensive structure collection dataset comprising 22,303 complexes.

Q10: At lines 377, "To mitigate data leakage, we excluded any similar protein pairs from the dataset used to train our model.": what's the definition of "similar protein pairs" here? What's the threshold used to define similarity? How was this threshold chosen?

Response: Thanks for your questions. The term "similar protein pairs" refers to pairs of proteins where both proteins individually exhibit significant sequence similarity to a protein pair in our test set. To clarify, we consider two proteins to be "similar" if their sequence identity exceeds a certain threshold. As we detailed in the "Similarity-based dataset splitting" part of the Methods section (and as we also discussed in our response to Q5), we empirically employed a sequence identity threshold of 50% (inspired from ProtGPT2 [1]). This means that if two proteins share more than 50% sequence identity, they are considered "similar". We have added a pointer in the manuscript to this section to make it easier for readers to follow the details of our data splitting process.

[1] Ferruz, N., Schmidt, S. & Höcker, B. ProtGPT2 is a deep unsupervised language model for protein design. *Nat Commun* **13**, 4348 (2022). <https://doi.org/10.1038/s41467-022-32007-7>

Q11: Model is predicting a predicted aligned error (PAE) score. What's the numerical definition of PAE? How are the model predicted PAE values evaluated? Related to the PAE definition, at lines 403-404, what does the sentence "With an average rank of 2.81 under 40 seeds" and Figure 3a middle mean?

Response: Thanks for your questions. Since the DockQ score is time consuming during training, we use the Root-Mean-Square Deviation (RMSD) between predicted and actual coordinates of protein2 as a training surrogate for PAE, which means the smaller the better. We have added a plot in the Supplementary Materials to illustrate the correlation between PAE and DockQ scores. For most of the examples, the PAE does show a negative correlation to the DockQ score.

For every PROTAC, we use RDKit to generate 40 different initial conformations by using 40 seeds, thus the model will predict 40 ternary structures. We rank all predictions according to their corresponding PAEs. The rank here and in the Figure 3a is the order of the first acceptable prediction (DockQ > 0.23). The average rank of 2.81 means that by average, an acceptable structure is within the top 3 predictions.

We have added more descriptions of the rank metric in the manuscript and pointed to the "Evaluation metrics" section of the Methods.

Minor Points

Q12: The line system of the paper get mis-aligned, but at roughly Lines 843 and 851, why is the PAE query denoted as "qVAE"? Shouldn't it be "qPAE"?

Response: Thanks for pointing out this mistake. You are right, they are supposed to be "qPAE", we have corrected them in the revised manuscript.

Q13: Some figures, e.g., Figures 3a, 3b, 3c, 6a, show some asterisks and notes, but it is not clear what they represent or what they point to. I'd suggest to maybe add a legend to the figure to clarify the meaning of these symbols.

Response: Thanks for the suggestion. The asterisks and notes are the statistical significance for different methods. We used independent t-test to calculate the probability of getting the observed results if there were actually no real effect (denoted as p-value). One asterisk indicates $p\text{-value} \leq 0.05$, which is considered statistically significant. Two and three asterisks indicate $p\text{-value} \leq 0.01$, and 0.001 respectively. We have included this annotation in the figure captions.

Q14: Figure 6a: what's the x axis? Why are data displayed in that way and not via, e.g., a whisker plot?

Response: Thanks for the question. The x-axis is the Decoder Type. The data are displayed as a swarm plot (similar to a violin plot) to show the distribution of DockQ scores for each decoder. We chose swarm plot because the box plot is hard to effectively represent the exact distribution. This swarm plot clearly illustrates how our proposed PPPD decoder enhances DockQ scores for certain samples, shifting them from unacceptable (< 0.23) to acceptable, medium, and even high quality, outperforming the decoder used in previous methods.

Q15: Figures 3a, 3b, 3c, and 3d can be greatly improved:

- The color scheme of the comparing methods is not consistent across the figures, nor it's their order.

Response: Thanks for the suggestion. Since different methods reported different criteria, each panel has different methods to compare. We have updated the plots to have consistent colors and the same relevant orders.

Revised:

Q16: Why aren't the comparing methods included in Figures 3d?

Response: Thanks for the suggestion. We have included other methods in Figure 3d.

Q17: Figure 3b and 3c report a "Method 4B", but I couldn't find a clear reference to what it stands for.

Response: Thanks for the question. We have added some method description to the main manuscript and included references to these methods.

Q18: I would suggest a more contrasting color palette for Figures 3f, 4c, 4d, and 4b.

Response: Thanks for the suggestion. These figures have been revised with a more contrasting color palette. Specifically, we now use orange and sky blue for ligand comparison, and gray and cyan for protein 2 comparison.

Revised:

Q19: Reference 32 is showing some "???" in its Bibliography entry.

Response: Thanks for pointing out the error. The "???" issue in reference 32 arose from the initial arXiv posting followed by publication in ICLR. We have corrected the reference in the manuscript.

Q20: The axis on Figure 6e is referred as "Seed Number", but it should be more like "Number of Sampled Random Conformations".

Response: Thanks for the suggestion. We have revised the axis title and manuscript to use "Number of Sampled Random Conformations" for a clear expression.

Q21: There is a typo in the Kabsch loss at line 956.

Response: Thanks for pointing out the problem. We have updated the manuscript to correct it to L_{kabsch_lig} .

Responds to Reviewer #1

Q1: The revised paper has addressed all of my concerns previously. I therefore recommend acceptance.

Response: We appreciate your positive feedback and constructive suggestions to improve our manuscript.

Responds to Reviewer #2

Q1: The code appears to be a usable resource with sufficient documentation for the users who want to reproduce the results from the paper. The README provides comprehensive instructions for installation and execution, and the availability of pre-trained models and datasets enhances reproducibility. However, there is a issue when I tried to access the TernaryDB dataset. The url link provided in the README file returns "Not Found". I further looked for the test.csv and val.csv in the data folder, but they are not there. This data availability issue might potentially make the result validation difficult.

Response: We apologize for the inconvenience caused by the incorrect link. We have uploaded the dataset to the release page; however, we didn't notice that the link becomes disabled after the release. We have updated the README to correct this link.

Responds to Reviewer #3

Q1: I thank the authors for a detailed evaluation of the comments, questions and suggestions. I observed significant improvements to the manuscript, but still encounter one major flaw that has not been properly addressed and one minor flaw that should be discussed. These points should be addressed before publication.

Major:

While I appreciate that the authors identified a validation set, the use of the validation set is not following best machine learning practices. Here are a few papers on the use of validation set and reason why not to use test set for hyperparameter optimization: Walsh et al. 2021 (<https://www.nature.com/articles/s41592-021-01205-4>) (Nature Methods), Chicco and Jurman 2022 (<https://doi.org/10.3389/fdata.2022.979465>), and Lones 2024 (<https://arxiv.org/pdf/2108.02497v5>) (specifically the section "Do avoid sequential overfitting"). In particular, the guide by Lones is helpful for academics to avoid common pitfalls.

Figure 1 should not show results on the test set, but rather the results on the validation set and the paragraphs "The architecture of DeepTernary" and "Effectiveness of model designs" should be written about the validation set instead of the test set. The authors noted that the model performing best on the validation set had similar hyperparameters to the model used to evaluate the test set. I would like to emphasize that this is the reverse of what is normally done in the field and that common practice should be adopted in future studies.

More concerning, the validation set was flagged as being similar to the test set. Therefore, there is data leakage from validation set to test set. Hence, the final results are not a true representation of how the model would generalize to novel PROTAC or MG cases.

Response: We sincerely appreciate your valuable feedback, which has been instrumental in enhancing our work. We fully agree with your suggestion to first perform hyperparameter optimization on the validation set before evaluating on the test set using the optimal parameters, and we will certainly adhere to this approach in future studies.

Regarding your concern about potential data leakage between the validation and test sets, we carefully addressed this by refining the validation set following the methodology of AlphaFold3 to ensure its dissimilarity to the test set. Specifically, we removed all complexes from the original validation set that exhibited a Tanimoto similarity greater than 0.85 to any complex in the test set. This process yielded a refined validation set comprising 11 PROTAC complexes and 137 MGD complexes. Hyperparameter optimization was then conducted solely on this updated validation set, and the revised results are presented in the following table. For PROTACs, since the Prompt-based Pocket Points Decoder (PPPD) was not used for pocket point extraction, there is no need to select the decoder type and head number, which are indicated by a placeholder (“-”). Notably, the best hyperparameters remain unchanged, implying that the main results and conclusions in our manuscript are still valid.

Supplementary Table A1 Results on the validation set for model selection. The highest VPS for each hyper-parameter is marked in bold.

	PROTAC			MG(D)
	$D_{\text{top-1}}$	D_{best}	VPS	VPS
IEGMN	-	-	-	8.91
PPPD	-	-	-	26.06
head1	-	-	-	26.06
head2	-	-	-	23.23
head4	-	-	-	22.81
head8	-	-	-	25.67
dim64	40.78	66.76	53.77	12.27
dim128	37.03	65.3	51.17	23.21
dim256	50.35	65.26	57.81	26.06
noise1	38.52	70.96	54.74	25.63
noise2	50.35	65.26	57.81	26.06
noise3	37.68	67.48	52.58	22.69

Results in the manuscript (Figure 1, Section “The architecture of DeepTernary” and “Effectiveness of Model Designs”) have been revised to reflect metrics on this new validation set.

To further assess the robustness of our method, we conducted 5-fold cross-validation experiments on the MG(D) training set for additional validation. The training set was randomly divided into five subsets, and in each iteration, the model was trained on four subsets and evaluated on the remaining one. The average performance across all folds was then calculated. The cross-validation results closely matched those obtained on our refined validation set, as shown below, further reinforcing the reliability of our evaluation approach.

Correlation between 5-fold cross-validation and the refined validation set on MG(D)s. Different point corresponds to different hyperparameter selection in Supplementary Table A1, with the x-axis showing the mean VPS scores from 5-fold cross-validation and the y-axis displaying those from the refined validation set.

We recognize that 5-fold cross-validation may not be the most suitable evaluation approach for PROTACs. Since our training set consists exclusively of non-PROTAC samples and given the practical challenges in obtaining their unbound structures, we could only rely on bound structures for validation. However, the inherent structural differences between non-PROTACs (being generally smaller and more rigid) and PROTACs resulted in consistently high DockQ scores during validation. These uniformly elevated scores unfortunately limit our ability to meaningfully assess hyperparameter impacts. So, we didn't include 5-fold cross-validation results on PROTACs.

Q2: In Q23 response to reviewers, the authors indicated that they re-docked monomer structures extracted from crystallized ternary complexes. This does not represent real-life case studies where the structure of monomers are not biased towards formation of the complex (e.g. optimal side-chains or loop conformations for protein-protein binding). Therefore, DeepTernary shows good performance in predicting complexes when starting from crystallized ternary complexes, but generalization to prospective cases where a crystal structure is not available is unclear. This should be addressed by testing DeepTernary with monomeric protein structures, or this limitation should be highlighted and discussed in the manuscript.

Response: We apologize for any confusion. In Q23 of our previous response letter, we were explaining the behavior in the training phase, where we disassembled and then re-docked crystallized ternary complexes. This approach was adopted because obtaining individual monomeric crystal structures for every training sample would have been prohibitively expensive. In contrast, during the inference phase, we followed the standard practice in PROTAC benchmarks and used unbound structures rather than crystallized ternary structures as input. Specifically, we utilize the corresponding monomer forms (unbound structures) of the two protein structures (E3 and POI), along with docked warheads and anchors from other PDB entries. For example, the PROTACs analyzed in Figure 6 of the BSA section lack available crystallized ternary structures, and the analysis was performed using predicted structures from unbound monomer structures as input.

We have now clarified this distinction between training and inference in the revised manuscript:

The architecture of DeepTernary

DeepTernary is designed to predict the structures of small molecule-induced ternary complexes, such as those formed by PROTACs and MG(D)s-induced E3 ligase with POI complexes. Unlike existing methods that rely on standard protein-protein docking programs to approximate the interaction between two proteins—often neglecting the presence of small molecules, DeepTernary employs a deep neural network to directly learn the intricate dynamics of protein-protein and protein-ligand interactions within ternary complexes. As shown in Fig. 1b, the model is trained by learning to re-dock disassembled ternary structures. During inference, for PROTAC-induced complexes, DeepTernary utilizes the respective monomer forms of the two protein structures (E3 ligase and target protein) along with docked warheads and anchors from other PDB entries (unbound structures), in addition to the PROTAC Simplified Molecular Input Line Entry System (SMILES) strings. For MG(D)-induced complexes, where obtaining unbound structures poses a challenge, we instead employ the in-complex form of the two protein structures (subjected to random rotations and transformations) and with the corresponding

287
288
289
290
291
292
293
294
295
296
297
298
299
300
301

Regarding MG(D)s, the absence of available unbound structures would necessitate substantial experimental effort to obtain them. Consequently, we utilized crystallized ternary complexes for input during both the training and inference phases. We have explicitly discussed this methodological limitation in the Discussion section.

719 present in the training set. Although we have collected a broad dataset from the PDB, there
720 is still room for improvement. Expanding the training data and incorporating lower-resolution
721 experimental datasets could further enhance the model's accuracy and applicability. Addition-
722 ally, for MG(D)s, we benchmarked using bound structures due to the substantial effort required
723 to obtain unbound structures. Updating the model to predict structure directly from sequence
724 can help address this limitation. We believe that future developments in this direction will
725 further extend DeepTernary's impact and enable broader application in drug discovery.

Q3: Lastly, in the spirit of academic congeniality, the authors may want to try the RDKit code below to keep both chemical handles rigid during conformational searches of a PROTAC: <codes are omitted>

Response: Thank you very much for sharing the RDKit code, which rigidly fixes PROTAC chemical handles during conformational searches. Implementing this strategy did improve the spatial alignment of warhead motifs in generated conformers, increasing the proportion of predictions with RMSD <math>< 2\text{\AA}</math> by 1 percentage point. We have updated the ligand pose accuracy section accordingly. For the three complexes shown in Figure 4f, the ligand pose accuracy improvement is significant, as shown below. We have acknowledged your help in the Acknowledgements Section.

Before:

4.38Å

4.36Å

8.33Å

After:

3.14Å

4.21Å

6.39Å

Responds to Reviewer #4

Q1: Thank you to the authors for the thorough revision of the manuscript. Happy to recommend it for publication.

Response: Thank you for your efforts in helping us to improve our manuscript.